

# The timing of the Ellesmerian Orogeny in Svalbard: A review

**Jean-Baptiste P. Koehl[1,2,3,4], John E. A. Marshall[5], Gilda Lopes[6]**

[1]Department of Geosciences, UiT The Arctic University of Norway in Tromsø, N-9037 Tromsø, Norway.

[2]CAGE – Centre for Arctic Gas Hydrate, Environment and Climate, NO-9037 Tromsø, Norway.

[3]Department of Geosciences, University of Oslo, P.O. Box 1047 Blindern, NO-0316 Oslo, Norway.

[4]Research Centre for Arctic Petroleum Exploration (ARCEx), University of Tromsø, N-9037 Tromsø, Norway.

[5]School of Ocean and Earth Science, University of Southampton, National Oceanography Centre, European Way, Southampton SO14 3ZH, UK.

[6]CIMA – Centre for Marine and Environmental Research, Universidade do Algarve, Campus de Gambelas, 8005-139, Faro, Portugal.

**Correspondence:** Jean-Baptiste P. Koehl (jeanbaptiste.koehl@gmail.com)

**Abstract**

In the Late Devonian–earliest Mississippian, Svalbard was affected by a short-lived episode of deformation named the Ellesmerian (Svalbardian) Orogeny. This event resulted in intense folding and thrusting in Devonian sedimentary successions. Deformation stopped prior to the deposition of Carboniferous–Permian sedimentary strata of the Billefjorden and Gipsdalen groups, which lie unconformably over folded Devonian strata. Later on, presumed Ellesmerian structures were reworked during Eurekan tectonism in the early Cenozoic and partly eroded. At present, record of Ellesmerian deformation is only preserved in narrow N–S-trending belts in central–northern, western and southern Spitsbergen. Despite extensive field studies, the timing of the Ellesmerian Orogeny is poorly constrained, and remains a matter of debate in places because of conflicting ages and because of the complex tectonic history of Svalbard. The present contribution aims at reviewing and discussing all available age constraints for Ellesmerian tectonism in Svalbard, which has great implications for the plate tectonic reconstructions of Arctic regions and for the tectonic history of Svalbard.

## 1. Introduction

The Ellesmerian Orogeny, also known as the Innuitian or Svalbardian Orogeny, refers to a short-lived episode of contraction–transpression that affected all levels of the crust and occurred in



the Late Devonian (–earliest Mississippian?) when parts of the tectonic plates now constituting most of the Arctic (Laurentia and Baltica) collided with each other and deformed Proterozoic to

mid-Paleozoic sedimentary basins and basement rocks in northeastern Russia (Malyshev et al., 2011; Luchitskaya et al., 2015), Canada (Trettin, 1973, 1991; Embry and Klovan, 1976; Embry, 1991; Harisson, 1995; Harisson and Brent, 2005; Piepjohn et al., 2008, 2013; Piepjohn and von Gosen, 2017) and Alaska (Grantz and May, 1984; Lane, 2007; Kumar et al., 2011), Proterozoic–Silurian metasedimentary rocks in northern and northeastern Greenland (Higgins et al., 2000;

Piepjohn et al., 2015), and Devonian collapse basins and Precambrian–lower Paleozoic basement in Norway (Roberts, 1983; Osmundsen et al., 1998) and Svalbard (Vogt, 1938; Harland et al., 1974; McCann, 2000; Piepjohn, 2000; Piepjohn et al., 2000; Figure 1).

  In Svalbard, Ellesmerian contraction (transpression?) followed the Caledonian Orogeny and led to the final accretion of Svalbard's three basement terranes (Harland and Wright, 1979;

Ohta et al., 1989, 1995; Harland et al., 1992; Gee and Page, 1994). Although early accounts envisioned hundreds–thousands of kilometer-scale strike-slip movements along N–S-striking faults like the Billefjorden and Lomfjorden fault zones (e.g., Harland et al., 1974, 1992), more recent studies have shown that such large scale strike-slip movements are unlikely (McCann, 2000; Piepjohn, 2000; Michalski et al., 2012). Evidence of Ellesmerian tectonism include dominantly

west-verging folds and thrusts within several kilometer-thick, Devonian, late–post-orogenic, collapse-related sedimentary rocks in central–northern Spitsbergen (Andrée Land Group including the Mimerdalen Subgroup; Vogt, 1938; Harland et al., 1974; Manby and Lyberis, 1992; Manby et al., 1994; Friend et al., 1997; Piepjohn and Dallmann, 2014; Dallmann and Piepjohn, 2020) and dominantly east-verging folds and thrusts in Devonian (–Middle Mississippian?) sedimentary

rocks in southern Spitsbergen (Marietoppen and Adriabukta formations; Dallmann, 1992; Bergh et al., 2011).

  Shortly after the end of Ellesmerian deformation, partly eroded Devonian sedimentary rocks in Spitsbergen were covered by fluvial, coal-rich deposits of the Billefjorden Group and shallow marine strata of the Gipsdalen Group deposited within narrow, kilometer- to tens of

kilometer-wide, N–S- to NW–SE-trending basins (Cutbill and Challinor, 1965; Maher Jr., 1996; McCann and Dallmann, 1996; Braathen et al., 2011; see Figure 2 for stratigraphy). Subsequently, Ellesmerian structures were reworked by Eurekan contraction–transpression during the opening of the Labrador Sea and Baffin Bay between Canada and Greenland (Chalmers and Pulvertaft, 2001;



Oakey and Chalmers, 2012), which resulted in the formation of the West Spitsbergen Fold-and-

Thrust Belt between Kongsfjorden and Sørkapp (Harland, 1969; Lowell, 1972; Harland and Horsfield, 1974; Maher et al., 1986; Dallmann et al., 1988, 1993; Andresen et al., 1994; Bergh and Grogan, 2003; see location in Figure 1) and of the Central Tertiary Basin in central Spitsbergen (Larsen, 1988; Petersen et al., 2016). As a result, Ellesmerian structures were overprinted and reworked and now commonly display the same trends, plunges, strikes, dips and kinematics as

Eurekan structures throughout the Arctic and, in many occurrences, coincide with and are indistinguishable from Eurekan structures (e.g., Birkenmajer, 1964; Piepjohn et al., 2007, 2008, 2013, 2015; Bergh et al., 2011; Piepjohn and von Gosen, 2017; Dallmann and Piepjohn, 2020).

At present, original Ellesmerian deformation is preserved only in a few narrow N–S-trending belts, including Dickson Land (Michaelsen et al., 1997; Piepjohn et al., 1997b;

Michaelsen, 1998; Piepjohn, 2000), Andrée Land (Dallmann and Piepjohn, 2020), and Blomstrandhalvøya (Thiedig and Manby, 1992; Buggisch et al., 1994; Figure 1). The best and most well-constrained example of Ellesmerian tectonism is observed in central–northern Spitsbergen, where folded Lower to lowermost Upper Devonian sedimentary rocks of the Andrée Land Group and Mimerdalen Subgroup are unconformably overlain by apparently undeformed uppermost

Devonian–lowermost Permian sedimentary strata of the Billefjorden and Gipsdalen groups (Vogt, 1938; Harland et al., 1974; Piepjohn, 2000; Piepjohn et al., 2000).

Recent U–Th–Pb geochronology on monazite grains yielded 373–355 Ma (latest Devonian–earliest Mississippian) ages for amphibolite facies metamorphism along a gently west-dipping shear zone in Prins Karls Forland (location in Figure 1) crosscutting Neoproterozoic

basement rocks. These data provide evidence and time constraints for Ellesmerian tectonism at depth of c. 15 kilometers (Faehnrich et al., 2017; Majka and Kośmińska, 2017; Schneider et al., 2018; Kośmińska et al., 2020). Potential Ellesmerian (greenschist) facies metamorphism and mylonitization was also potentially identified in Oscar II Land (location in Figure 1) and dated to 365–344 Ma through $^{40}Ar$–$^{39}Ar$ and U–Th–Pb geochronology (Barnes et al., 2020).

Nonetheless, despite extensive previous works, Ellesmerian tectonism at shallow crustal levels lacks accurate time constraints and, in places, it is possible that structures ascribed to this event may have formed during the early Paleozoic Caledonian Orogeny or during the early Cenozoic Eurekan tectonic event (Rippington et al., 2010). For example, east- to northeast-plunging folds trending parallel to the inferred late–post-orogenic extension direction in Middle



Devonian collapse basins in western Norway were initially interpreted as Late Devonian–
Mississippian, Ellesmerian contractional–transpressional structures (Roberts, 1983). These are
now known to have formed as transtensional folds during extensional collapse of the Caledonides
(Chauvet and Séranne, 1994; Osmundsen and Andersen, 1994; Fossen et al., 2013). Thus, it is
paramount to carefully constrain the timing of Ellesmerian deformation throughout the Arctic to
be able to evaluate the extent and impact of this tectonic event from a regional perspective.

Thus far, Ellesmerian deformation is thought to have initiated in the Late Devonian–Early
Mississippian, possibly in the late Frasnian–Famennian (Vigran, 1964; Allen, 1965, 1973; Pcelina
et al., 1986; Brinkmann, 1997; Schweitzer, 1999; Piepjohn et al., 2000; Figure 2). The onset of
deformation was presumably recorded by the deposition and syn-depositional deformation of
coarse-grained sedimentary rocks of the Mimerdalen Subgroup in the late Famennian
(Planteryggen and Plantekløfta formations in Figure 2; Piepjohn and Dallmann, 2014).
Deformation is believed to have stopped prior to the deposition of sedimentary rocks of the
Billefjorden Group in the late Tournaisian (Vogt, 1938; Piepjohn, 2000).

The present contribution focuses on the debate around the timing of Ellesmerian tectonism
throughout Spitsbergen. In northern Spitsbergen, Ellesmerian deformation was constrained to the
late Famennian–earliest Mississippian by the identification of one specimen of *Retispora
lepidophyta* in folded rocks of the Plantekløfta Formation (Schweitzer, 1999; Piepjohn et al., 2000).
However, recent palynological and paleontological studies in northern–central Spitsbergen suggest
slightly revised ages for the stratigraphic units used to constrain the timing of the Ellesmerian
Orogeny, including a Middle Devonian (minimum upper Givetian) age for rocks of the Tordalen
Formation (Mimerdalen Subgroup; Berry and Marshall, 2015; Newman et al., 2019) and a mid
Famennian age for the base of the Billefjorden Group (Scheibner et al., 2012; Lindemann et al.,
2013; Marshall et al., 2015; Gilda M. Lopes pers. obs., 2019). In addition, the timing of Ellesmerian
deformation varies somewhat from north to south in Spitsbergen, and study of a palynological
assemblage in the Adriabukta Formation in southern Spitsbergen constrained Ellesmerian folding
and faulting to the Viséan (Middle Mississippian; Birkenmajer and Turnau, 1962; Figure 2). The
present contribution reviews time constraints for Ellesmerian tectonism in central–northern,
southern, and western Svalbard and briefly discusses their implications.

Constraining the timing of the Ellesmerian Orogeny in Svalbard with accuracy is of
importance for paleogeographic and plate tectonics reconstructions in the Arctic. It is also





important for the tectonic history of Svalbard, e.g., to evaluate potential interplay between late–post-Caledonian extensional collapse, which resulted in the deposition of several kilometers thick collapse basins (e.g., Murascov and Mokin, 1979; Manby and Lyberis, 1992; Friend et al., 1997; Braathen et al., 2018), and contractional tectonic processes that resulted in intense folding of these

deposits (Vogt, 1938; Piepjohn, 2000; Dallmann and Piepjohn, 2020). Furthermore, the present study has implications for the methods used by geologists to interpret tectonic events worldwide.

**Review of age constraints in northern and central Spitsbergen**

*Age of the Mimerdalen Subgroup*

The identification of one specimen of *Retispora lepidophyta* within strata of the Plantekløfta Formation by Brinkmann (1997, his table 14.3) and Piepjohn et al. (2000; published in Schweitzer, 1999, plate 6 in their figure 10 and plate 7 in their figure 1) suggests a late Famennian age for the top fo the Plantekløfta Foramtion and, hence, that Ellesmerian tectonism terminated during the Famennian–Tournaisian in northern and central Spitsbergen.

Recent studies clearly demonstrated that the interpretation of *Retispora lepidophyta* by Brinkmann (1997), Schweitzer (1999) and Piepjohn et al. (2000) is erroneous. Notably, the lone figured specimen interpreted as *Retispora lepidophyta* by Schweitzer (1999) and Piepjohn et al. (2000) differs in size and shows significantly different morphological structures from typical *Retispora lepidophyta* (Playford, 1976; Berry and Marshall, 2015, their supplement DR3). In

addition the fovea that characterise the spore's exoexine appear to be the result of damage by cubic diagenetic pyrite. Attempts have been made to locate the *Retispora lepidophyta* specimen figured in Brinkmann (1997) and Schweitzer (1999) and used by Piepjohn et al. (2000) to date Ellesmerian tectonism in central Spitsbergen for further analysis. These attempts were unfortunately unsuccessful (John E. A. Marshall pers. obs., 2020).

Berry and Marshall (2015) re-evaluated the age of the Plantekløfta Formation to be early Frasnian based on fossils and miospores (ca. 383–380 Ma; see also their supplements; Figure 2). In addition, the paleontological study of Newman et al. (2019, 2020, 2021) recorded the presence of articulated fish in the Fiskekløfta Member of the Tordalen Formation (Figure 2), i.e., undoubtedly *in situ* fossils, demonstrating a late–latest Givetian (ca. 385–383 Ma; Middle

Devonian) age for this stratigraphic unit instead of late Famennian. If the relatively coarser grain-size of the sedimentary deposits of the Plateryggen and Plantekløfta formations indeed reflects the



onset of Ellesmerian tectonism as suggested by Piepjohn and Dallmann (2014), the new paleontological–palynological ages constrain the initial phase of the Ellesmerian Orogeny at 383–380 Ma.

A late Famennian age for the Plantekløfta Formation based on the lone specimen of *Retispora lepidophyta* in central Spitsbergen is the only contradictory evidence against a mid Famennian age for the base of the Billefjorden Group and older age for the Mimerdalen Subgroup (Scheibner et al., 2012; Lindeman et al., 2013; Berry and Marshall, 2015; Marshall et al., 2015; Lopes et al., 2019; Newman et al., 2019; Gilda M. Lopes pers. obs., 2019).


### Age of the Billefjorden Group

Recent palynological studies in central Spitsbergen dated the base of the Billefjorden Group in Triungen (see Figure 1 for location) to the mid Famennian (maximum ca. 365 Ma; Lindemann et al., 2013; Marshall et al., 2015; Gilda M. Lopes pers. obs., 2019; Figure 2). At least 30 samples

contained characteristic Famennian spore assemblages including *Cyrtospora cristifer, Cornispora monocornata, Cornispora bicornata, Cornispora tricornata, Lophozonotriletes lebedianensis*, *Knoxisporites dedaleus*, *Grandispora gracilis*, *Spelaeotriletes papulosus*, *Cristatisporites lupinovitchi*, *Lagenosisporites* sp., *Grandispora famensis* and *Tergobulasporites immensus* (Marshall et al., 2015). Some samples from the lower part of the Billefjorden Group in Billefjorden

also contained *Retispora lepidophyta* (Gilda M. Lopes obs. comm., 2019). These spore assemblages were also identified in sedimentary rocks in the lower part of the Billefjorden Group in northeastern Spitsbergen (Scheibner et al, 2012), thus strengthening a Famennian age for the base of this stratigraphic unit throughout northern and central Spitsbergen. Note that the base of the Billefjorden Group in Bjørnøya also was also dated as Famennian based on palynology (Kaiser,

1970; Worsley and Edwards, 1976; Lopes et al., 2021). This strongly suggests that the Ellesmerian deformation, which ended prior to the deposition of the Billefjorden Group (Piepjohn, 2000), must have been terminated by the mid Famennian in central–northern Spitsbergen. This implies a maximum duration of 18 Ma for this tectonic event.

Piepjohn and Dallmann (2014) proposed that the mid–late Famennian spores identified in

the lower part of the Billefjorden Group were reworked based on their identification of one specimen of *Retispora lepidophyta* within the Plantekløfta Formation (Piepjohn et al., 2000). However, since this specimen clearly is a misidentification (Berry and Marshall, 2015, their



supplement DR3), the claim of reworking of mid–late Famennian spores found within the base of the Billefjorden Group in Triungen is no longer valid.


### *Other time constraints for deformation in central–northern Spitsbergen*

At least some of the deformation in Lower to lowermost Upper Devonian strata of the Andrée Land Group and Mimerdalen Subgroup in central and northern Spitsbergen is early Cenozoic in age because uppermost Devonian–Mississippian strata of the Billefjorden Group,

which overlie the Andrée Land Group and Mimerdalen Subgroup in the area, are intensely sheared top-west, e.g., in Pyramiden (Koehl, 2021) and Garmdalen (Koehl et al., 2020, 2022 submitted; locations in Figure 1). This is further supported by the interpretation of seismic data adjacent nearshore portions of Billefjorden showing the presence of a bedding-parallel décollement between the Wood Bay Formation and the Gipsdalen Group (Koehl et al., 2020; Koehl et al., 2022 in prep.).

These suggest a significant impact of strain partitioning during Eurekan deformation. Eurekan strain partitioning is further illustated by tight plastic folding of Lower Devonian strata of the Andrée Land Group and brittle brecciation of the unconformity with Upper Pennsylvanian–Permian strata in Yggdrasilkampen (Manby et al., 1994 their figure 11), and by décollements within Middle Devonian deposits near the Billefjorden Fault Zone in Wijdefjorden (John E. A.

Marshall pers. obs., 2022; see Figure 1 for location).

Another argument corroborating these data is the involvement in folding of Carboniferous picritic dykes dated at ca. 357 Ma (Evdokimov et al., 2006; monchiquite dykes in Gayer et al., 1966 and Manby and Lyberis, 1996) intruding Lower Devonian sedimentary rocks at Krosspynten (see location in Figure 1).

In addition, part of the deformation recorded by Lower to lowermost Upper Devonian strata of the Andrée Land Group and Mimerdalen Subgroup is possibly related to extensional detachment folding in the Devonian (Chorowicz, 1992; Roy, 2007, 2009; Roy et al., unpublished). This is also supported by recent field and geochronological studies in northwestern Spitsbergen (Braathen et al., 2018, 2020).

Thus, it is unclear how much (if any at all) of the deformation observed within Lower to lowermost Upper Devonian strata of the Andrée Land Group and Mimerdalen Subgroup in central–northern Spitsbergen actually reflects Ellesmerian tectonism.





**Review of age constraints in southern Spitsbergen**

*Age of the Adriabukta Formation*

In southern Spitsbergen, Lower–Middle Devonian sedimentary rocks of the Marietoppen Formation (time equivalent to the Pragian–Eifelian Wood Bay and Grey Hoek formations of the Andrée Land Group in central–northern Spitsbergen; Figure 2) unconformably overlie Precambrian–early Paleozoic basement rocks and are overlain by sedimentary strata of the Adriabukta Formation that were deformed into tight east-verging folds presumably during Ellesmerian tectonism. The age of the Adriabukta Formation was dated to the Middle Mississippian through analysis of palynomorphs from black shales at the base and within the Formation (Birkenmajer and Turnau, 1962; Figure 2). Dallmann et al. (1999) noted that because of the age discrepancy between the Middle Mississippian Adriabukta Formation and the Lower to lowermost Upper Devonian Andrée Land Group in central–northern Spitsbergen, the folding of the Adriabukta Formation could not be correlated to Svalbardian folding. Nevertheless, in 2011, W. Dallmann suggested that the Adriabukta Formation is actually Late Devonian in age based on structural correlation between presumed Ellesmerian structures in the Adriabukta Formation and Ellesmerian fold-and-thrust belts in central–northern Spitsbergen, thus generating a discussionebate around the actual age of the formation. This is referenced as "W. Dallmann pers. comm. 2009" in Bergh et al. (2011).

The discussion initiated by W. Dallmann around the age of the Adriabukta Formation is neither based on published material nor on specific scientific evidence. By contrast, Birkenmajer and Turnau (1962) identified a count of 350 spore specimens from the Adriabukta Formation including specimens of *Lycospora*, *Tripartites* and *Triquitrites*, which were then and are still characteristic of the Middle Mississippian (Hughes and Playford, 1961; Playford, 1962, 1963; Clayton 1996). Later palynological studies in Svalbard (Billefjorden; Lopes et al., 2019) and Europe (Clayton et al., 1977) corroborate the Middle Mississippian ages obtained by Birkenmajer and Turnau (1962) for the Adriabukta Formation. Thus, the speculationdebate around the Middle Mississippian ages obtained for the Adriabukta Formation by Birkenmajer and Turnau (1962) is not justified and a Middle Mississippian age is entirely justified. The Adriabukta Formation in southern Spitsbergen is therefore a time-equivalent of the Billefjorden Group (e.g., Lopes et al., 2019).





The Middle Mississippian age of the Adriabukta Formation suggests that folding within
this stratigraphic unit cannot be Late Devonian and is therefore not related to Ellesmerian
tectonism. A more likely origin for deformation within the Adriabukta Formation is the early
Cenozoic Eurekan tectonic event. The tightly folded character of the Adriabukta Formation was
previously proposed to be related to the dominance of weak shale and to Cenozoic strain
partitioning by Birkenmajer and Turnau (1962). This scenario is now the most likely explanation
for differential deformation of shales of the Adriabukta Formation and for folding of the
Marietoppen Formation in southern Spitsbergen.

### Age of the Hornsundneset Formation

In Hornsundneset (Figure 1), Siedlecki and Turnau (1964) analyzed eight samples from the
Hornsundneset and Sergeijevfjellet formations of the Billefjorden Group. They proposed a
Serpukhovian (Late Mississippian) age based on palynological results. However, a re-evaluation
of their results showed that the Billefjorden Group in Hornsundneset (location in Figure 1) is
Middle Mississippian in age (Dallmann et al., 1999; Krajewski and Stempien-Salek, 2003), i.e.,
contemporaneous with the Adriabukta Formation (Figure 2).

Interestingly, the Hornsundneset and Sergeijevfjellet formations are dominated by
relatively hard, flat-lying beds of sandstone. Though located closer to the early Cenozoic collision
zone with Greenland (i.e., within the West Spitsbergen Fold-and-Thrust Belt), these formations are
relatively undeformed compared to the shale-dominated Adriabukta Formation (Siedlecki, 1960).
This further supports a significant impact of strain partitioning on deformation patterns during the
Eurekan tectonic event in southern Spitsbergen.

### Other time constraints

In Adriabukta (location in Figure 1), the Adriabukta Formation is truncated by a major
shear zone, the Mariekammen Shear Zone, which comprises hundreds of meter-long lenses of
Cambrian metasedimentary basement rocks, shows a top-east reverse sense of shear, and is
unconformably overlain upwards by mildly folded Pennsylvanian strata of the Gipsdalen Group
(Hyrnefjellet Formation), thus possibly reflecting Ellesmerian tectonism (Birkenmajer and Turnau,
1962; Birkenmajer, 1964; Dallmann, 1992; Bergh et al., 2011). However, these previous studies
did not account for the impact of Eurekan tectonism in southern Spitsbergen. A simple restoration





of the shear zone prior to Eurekan deformation shows that, if this structure is indeed Mississippian in age, it must have formed as a normal fault and therefore cannot reflect Ellesmerian contractional deformation (Supplement S1). It should be noted that other workers proposed that the Mariekammen Shear Zone formed as an early Cenozoic structure (Dallmann, 1992; von Gosen and Piepjohn, 2001).

The Adriabukta Formation was intruded by two, thin, bedding-parallel, Early Cretaceous dolerite sills of the Diabasodden Suite (Senger et al., 2013) that are folded together with bedding surfaces (Birkenmajer and Morawski, 1960; Birkenmajer, 1964). If the Adriabukta Formation was already folded in the Early Cretaceous, the sills would have truncated both fold structures and bedding surfaces. For sills to intrude along bedding surfaces, these must have remained relatively

undeformed, sub-planar, and sub-horizontal until the Early Cretaceous. The Early Cretaceous sills and Middle Mississippian sedimentary strata were then folded together during subsequent Eurekan deformation. The two Early Cretaceous sills therefore further constrain the age of folding within the Adriabukta Formation and Marietoppen Formation to the early Cenozoic.

An early Cenozoic age for folding of shales of the Adriabukta Formation is further

suggested by similar tight, east-verging fold geometries in Lower Triassic sedimentary strata incorporated as lenses into basement rocks in Fiskeknatten (locaton shown in Figure 1; Birkenmajer, 1964).

Ellesmerian deformation may be recorded in southernmost Spitsbergen (Røkensåta; Figure 1 for location) where two outcrops of limited geographical extent (<< one km$^2$) show poorly

exposed, gently dipping, shale-rich, Lower Triassic sedimentary rocks over folded Middle Devonian strata (Dallmann, 1992). However, the two outcrops are of small size because extensively eroded and the stratigraphic contact between Devonian and Triassic rocks is completely covered by loose material and located on steep mountain flanks (i.e., inaccessible for detailed inspection). In addition, Triassic successions in Spitsbergen dominantly consist of shale (Worsley and Mørk,

1978), and strain partitioning during early Cenozoic contraction is now known to have had a considerable influence on the deformation of shale units in southern Spitsbergen (e.g., tightly folded Middle Mississippian Adriabukta Formation versus undeformed Middle Mississippian Hornsundneset Formation; Siedlecki, 1960; Birkenmajer and Turnau, 1962). Furthermore, folds within Middle Devonian rocks in Røkensåta appear to die out upwards (see figure 4a in Dallmann,

1992). It is therefore possible that deformation in Røkensåta is also early Cenozoic in age.



Such heavily eroded and limited outcrops need to be interpreted with extreme caution. Lower Triassic strata throughout Spitsbergen are well known for hosting bedding-parallel Eurekan décollements (Maher, 1984; Maher et al., 1986, 1989; Andresen et al., 1988; Bergh and Andresen, 1990; Haremo and Andresen, 1992; Andresen et al., 1992; Dallmann et al., 1993; Bergh et al.,

1997). The most spectacular examples include the décollement in dark shales on the Midterhuken Peninsula (Maher, 1984; Dallmann et al., 1993; location shown in Figure 1) the Berzeliustinden thrust in southern Spitsbergen (Dallmann, 1988), the Triassic décollement penetrated by the 7816/12-1 exploration well and well imaged on seismic data in Reindalspasset (Eide et al., 1991; Koehl, 2021 his figure 5g; see Figure 1 for location), and the "Lower Décollement Zone" in eastern

Spitsbergen (Andresen et al., 1992; Haremo and Andresen, 1992). A similar structure may very well have decoupled Eurekan deformation between folded Middle Devonian and overlying gently dipping Lower Triassic sedimentary strata in Røkensåta. This example stresses the importance of detailed inspection of extensively eroded outcrops, especially in glaciated Arctic areas, and highlights potential flaws in long-distance interpretation of kilometer-scale mountain flanks.


**Review of age constraints in western Spitsbergen**

***Conodont age in Blomstrandhalvøya***

In western Spitsbergen, Thiedig and Manby (1992) and Kempe et al. (1997) showed that west-verging thrusts crosscut Proterozoic and Devonian sedimentary rocks in Blomstrandhalvøya

(location in Figure 1). They used the westwards transport direction of these thrusts to suggest that they record Ellesmerian tectonism because it is comparable to observations along inferred Ellesmerian thrusts in Dickson Land and Andrée Land in central–northern Spitsbergen (Vogt, 1938; Harland et al., 1974; Piepjohn, 2000).

In addition, Kempe et al. (1997) also noted the presence of small NW-verging thrusts on

Blomstrandhalvøya. Notably, they argued that the size of these thrust was different from that of Ellesmerian structures and concluded that they must therefore be post-Devonian. Kempe et al. (1997) argued that, even though the NW-verging thrusts seemed to have formed in the early Cenozoic, NW-directed transport directions are not typical of early Cenozoic Eurekan tectonism, which produced NE-verging thrusts and folds in adjacent areas of Brøggerhalvøya (Bergh et al.,

2000; Piepjohn et al., 2001; see location in Figure 1). They therefore proposed that NW-verging thrusts on Blomstrandhalvøya formed during a discrete tectonic event in the Pennsylvanian–



Cretaceous. However, such a tectonic event is, thus far, unheard of in Spitsbergen. It is therefore more likely that the NW-verging thrusts in Blomstrandhalvøya formed in the early Cenozoic.

In western Blomstrandhalvøya, one sample in a presumably undeformed karst infill within a few meters wide fissure in Proterozoic basement marbles yielded a Pennsylvanian–Permian age based on conodont fauna (Buggisch et al., 1994; Figure 2). Since the karst infill was apparently not deformed, Buggisch et al. (1994) argued that the conodont fauna potentially constrained the formation of folds and west-verging thrusts on Blomstrandhalvøya to the Late Devonian (Ellesmerian).

Nevertheless, several aspects of this feature call for caution regarding its bearing for Ellesmerian tectonism. First, despite being located in an area strongly deformed by Eurekan tectonism, e.g., Blomstrandhalvøya (e.g., NW-verging thrusts of Kempe et al., 1997) and Brøggerhalvøya; (Bergh et al., 2000; Piepjohn et al., 2001; Figure 1), the Pennsylvanian–Permian cave seems to have escaped early Cenozoic deformation. This is possibly due to partitioning of

Eurekan strain, which is known to have had a significant influence on deformation patterns in Brøggerhalvøya (e.g., Bergh et al., 2000). Thus, this small-scale karst feature is not an appropriate marker to discuss the timing of regional tectonic events in Blomstrandhalvøya.

        Second, the cave is located within relatively undeformed Proterozoic marbles, and away from presumed Ellesmerian west-verging thrusts and associated deformed Lower Devonian

sedimentary rocks on Blomstrandhalvøya. Hence, the karst infill is inappropriate to constrain the timing of Ellesmerian deformation in Blomstrandhalvøya. The deformation in basement marble (if any at all at the location of the karst) could very well be Caledonian as previously suggested by Michalski (2018).

        Third, the karst is the only one of its kind yielding a Pennsylvanian–Permian age and is, moreover, based on only one sample with a poorly preserved conodont fauna (Buggisch et al.,

1994). In their study, Buggisch et al. (1994) specified that the assignation to published species was difficult due to the poor preservation of the elements. Hence, further studies of caves and conodont fauna on Blomstrandhalvøya are therefore needed to further assess the reliability of the age obtained by Buggisch et al. (1994) and its implication (if any at all) for Ellesmerian tectonism.

Considering all pieces of evidence gathered thus far, the folds and thrusts in Proterozoic–Lower Devonian rocks in Blomstrandhalvøya may all be Caledonian and Eurekan in age since no appropriate constraints are available to date any potential Ellesmerian deformation.



### *Amphibolite facies metamorphism in Prins Karls Forland*

In Prins Karls Forland (see Figure 1 for location), amphibolite facies metamorphism was dated to 373–355 Ma by ion microprobe and $^{40}Ar$–$^{39}Ar$ geochronology, and was postulated to be prograde and, thus, to record deep-crustal Ellesmerian tectonism (c. 15 kilometers depth; Majka and Kośmińska, 2017; Faehnrich et al., 2017; Schneider et al., 2018; Kośmińska et al., 2020). This episode of deep-crustal metamorphism is coeval with shallow-crustal Ellesmerian tectonism in

central–northern Spitsbergen dated to ca. 383–365 Ma by recent paleontological and palynological studies (Scheibner et al., 2012; Lindemann et al., 2013; Marshall et al., 2015; Berry and Marshall, 2015; Newman et al., 2019; Gilda M. Lopes pers. obs., 2019).

     However, kinematic indicators along the dated shear zone display top-SW to top-NW normal sense of shear (Schneider et al., 2018 their figure 3b, e and f), which is incompatible with

a formation during contractional (Ellesmerian) tectonism. Instead, the shear sense rather suggests a close relationship with Devonian extensional collapse of the Caledonides. Notably, amphibolite-facies metamorphism in Prins Karls Forland is also coeval with and occurred at comparable depth as deep-crustal, late Caledonian, high-pressure metamorphism along the conjugate eastern–northeastern Greenland margin (Gilotti et al., 2004; McClelland et al., 2006; Augland et al., 2010,

2011), which developed synchronously with the deposition of Devonian–Mississippian collapse basins along low-angle extensional detachments at the surface (Stemmerik et al., 1991, 1998, 2000; Larsen and Bengaard, 1991; Strachan, 1994; Larsen et al., 2008). During late–post-orogenic collapse, deep contractional tectonics occurring typically at greenschist–amphibolite-facies conditions (Snoke, 1980; Lister and Davis, 1989; Krabbendam and Dewey, 1998) are commonly

associated with near-surface extension (Platt, 1986; Rey et al., 2001, 2011; Teyssier et al., 2005).

     Amphibolite-facies metamorphism in Prins Karls Forland was also coeval with collapse-related core complex exhumation in northwestern Spitsbergen (latest movement at 368 Ma; Braathen et al., 2018). Hence, despite the postulated prograde character of amphibolite-facies metamorphism in Prins Karls Forland, its timing appears to coincide with Late Devonian

extensional events in nearby areas. If the postulated prograde character of amphibolite-facies metamorphism in Prins Karls Forland is to be reconciled with the observed overall top-SW to top-NW normal sense of shear (Schneider et al., 2018 their figure 3b, e and f) and with extensional tectonics in northwestern Spitsbergen (Braathen et al., 2018), then the shear zone and associated



prograde metamorphism may reflect gradual burial linked to the deposition of thick collapse
sediments and/or normal movements along the shear zone.

The geochronological ages obtained by Kośmińska et al. (2020) show broad ranges (430–
336 Ma for monazite population I, 419–261 Ma for population II, and 443–226 Ma for population
III) all ranging from the Silurian (Caledonian?) to the Carboniferous–Triassic. In addition, the ages
obtained are associated with large $\sigma_1$ errors (12.4–20.2 Myr for population I, 19.6–49.9 Myr for
population II, and 17.1–64.4 Myr for population III; see online supplement S1 in Kośmińska et al.,
2020). Since the length of Ellesmerian tectonism in shallow-crustal Lower to lowermost Upper
Devonian sedimentary rock in central–northern Spitsbergen is constrained to a maximum time span
of 18 million years (383–365 Ma), i.e., a time span comparable with the $\sigma_1$ errors associated with
the ages obtained by Kośmińska et al. (2020), these ages are inappropriate to discuss the timing of
Ellesmerian tectonism in Svalbard (Schaltegger et al., 2015).

Furthermore, since the Late Devonian–Mississippian (373–355 Ma) amphibolite-facies
metamorphism in basement rocks in Prins Karls Forland probably occurred at c. 15 kilometers
depth, the timing and nature of metamorphism may not have any implications for the nature of
paleostress and resulting deformation in shallow-crustal Devonian sedimentary rocks in
Spitsbergen (e.g., coeval ultra-high pressure metamorphism at depth and extensional collapse at
the surface in Greenland in the Devonian–Mississippian; Strachan, 1994; Gilotti et al., 2004;
McClelland et al., 2006).

*Greenschist facies metamorphism and thermal overprints in Oscar II Land*

In Oscar II Land (location in Figure 1), greenschist facies metamorphism yielded 365–344
Ma [40]Ar–[39]Ar and U–Th–Pb ages (Barnes et al., 2020) suggesting it potentially reflects Ellesmerian
deformation. However, these ages were re-evaluated to ca. 410 Ma (Early Devonian; Ziemniak et
al., 2020). This episode of low-grade metamorphism was therefore coeval with the deposition of
Lower Devonian sedimentary rocks in central and northern Spitsbergen in the Devonian Graben
during late–post-orogenic collapse of the Caledonides and, thus, is most likely related to
extensional processes (Gee and Moody-Stuart, 1966; Friend et al., 1966; Friend and Moody-Stuart,
1972; Murascov and Mokin, 1979; Manby and Lyberis, 1992; Friend et al., 1997; McCann, 2000).

In addition, Michalski et al. (2017) evidenced two episodes of thermal overprints at 377–
326 and ca. 300 Ma in pre-Caledonian rocks in Oscar II Land using [40]Ar–[39]Ar geochronology. The





435    latter event is believed to be related to rifting. The former event at 377–326 Ma partly overlaps

with the presumed timing of the Ellesmerian Orogeny in central–northern Spitsbergen at ca. 383–

365 Ma (Scheibner et al., 2012; Lindemann et al., 2013; Marshall et al., 2015; Berry and Marshall,

2015; Newman et al., 2019; Gilda M. Lopes pers. obs., 2019) and with the timing of 373–355 Ma

amphibolite facies metamorphism in western Spitsbergen (Majka and Kośmińska, 2017; Faehnrich

et al., 2017; Schneider et al., 2018; Kośmińska et al., 2020). It is, however, not possible to infer

tectonic stress orientation and this event may very well be related to Ellesmerian tectonism or to

late Caledonian extensional processes in northeastern Greenland and Prins Karls Forland

(Stemmerik et al., 1991, 1998, 2000; Larsen and Bengaard, 1991; Strachan, 1994; Larsen et al.,

2008; Schneider et al., 2018; see also previous section) and in northern Spitsbergen (Chorowicz,

1992; Roy, 2007, 2009; Braathen et al., 2018, 2020; Roy et al., unpublished).

**Discussion and re-evaluation of the timing and extent of Ellesmerian tectonism**

        The present brief review of age constraints in Spitsbergen shows a few noteworthy aspects

of dating Ellesmerian tectonism in Svalbard. In southern Spitsbergen, Middle Mississippian

palynological ages for the tightly folded, shale-rich Adriabukta formation (Birkenmajer and

Turnau, 1962) and its intrusion by two Early Cretaceous dolerite sills that are folded together with

bedding surfaces (Birkenmajer and Morawski, 1960; Birkenmajer, 1964) show that folding in this

area may be exclusively and entirely early Cenozoic in age. Comparable Middle Mississippian

palynological ages for the contemporaneous but undeformed, sandstone-dominated Hornsundneset

Formation c. 20 kilometers to the southwest (Siedlecki, 1960; Siedlecki and Turnau, 1964) and

mild folding of clastic-rich Pennsylvanian–Permian rocks in Adriabukta (Birkenmajer, 1964;

Bergh et al., 2011) illustrate the strong impact of Eurekan strain partitioning on deformation

patterns in southern Spitsbergen as previously considered by Birkenmajer and Turnau (1962) and

Koehl (2020a).

The only possible record of Ellesmerian tectonism in southern Spitsbergen occurs at

Røkensåta. However, as previously discussed, the low quality of the only two exposures

(stratigraphic contact covered by loose material), their very limited extent ($<<$ one km$^2$), their

inaccessibility for detailed inspection (located on steep mountain flanks), the significant impact of

early Cenozoic strain partitioning in southern Spitsbergen (Birkenmajer and Turnau, 1962), and

the geometry of folds within Middle Devonian rocks at this locality (dying out upwards; Dallmann,



1992) call for caution and further detailed investigation of structural and stratigraphic relationships at this locality. Nevertheless, if Eurekan tectonism alone produced the intense deformation in Adriabukta, it is possible that deformation in Røkensåta is exclusively early Cenozoic as well.

In central–northern Spitsbergen, Ellesmerian tectonism was constrained to ca. 383–365 Ma (i.e., a maximum duration of 18 million years) by recent paleontological and palynological studies in sedimentary rocks of the Mimerdalen Subgroup (Berry and Marshall, 2015; Newman et al., 2019, 2020, 2021) and Billefjorden Group (Scheibner et al., 2012; Lindemann et al., 2013; Marshall et al., 2015; Gilda M. Lopes pers. obs., 2019). The only contradictory late Famennian age obtained by Piepjohn et al. (2000) via identification of one specimen of *Retispora lepidophyta* in one sample 475 of the Plantekløfta Formation of the Mimerdalen Subgroup is now known to be a clear misidentification (Berry and Marshall, 2015 their supplement DR3).

        Despite the accurate paleontological–palynological time constraints for Ellesmerian tectonism in central–northern Spitsbergen, no geochronological constraints exist yet for discrete Ellesmerian structures. In addition, the central–northern Spitsbergen area was strongly affected by 480 early Cenozoic Eurekan tectonism during which strain partitioning played an important role in localizing deformation in weak, shale-rich lithostratigraphic units like the Billefjorden Group (e.g., Koehl, 2021). Moreover, evidence for extensional detachment-related folding in northerwestern (Braathen et al., 2018, 2020) and northern Spitsbergen (Chorowicz, 1992; Roy, 2007, 2009; Roy et al., unpublished) in Middle–Late Devonian may also have contributed to deformation patterns 485 observed within Lower to lowermost Upper Devonian strata of the Andrée Land Group and Mimerdalen Subgroup. Thus, it is unclear how much (if any at all) of the deformation observed within Lower to lowermost Upper Devonian strata in central–northern Spitsbergen actually reflects Ellesmerian tectonism. Further studies are therefore clearly needed to quantify the impact of the Ellesmerian Orogeny and to segregate discrete Ellesmerian from Devonian extensional 490 (detachment) faulting and folding and from early Cenozoic Eurekan folding and thrusting.

        Another line of controversy is the increadibly rapid switch from extension-related normal faulting in the Early–Middle Devonian to Ellesmerian contraction in the Late Devonian, and back to dominantly extensional setting in the mid Famennian in central–northern Spitsbergen. Notably, the Wood Bay Formation and Fiskekløfta Member of the Tordalen Formation are downfaulted by 495 normal faults in southern Hugindalen and unconformably covered by the Planteryggen Formation (Hugindalen Phase in Piepjohn, 2000 and Dallmann and Piepjohn, 2020). The Fiskekløfta Member





was dated to the latest Givetian (top of the unit at ca. 383 Ma) and the Plantekløfta Formation to the early Frasnian (383–380 Ma; Berry and Marshall, 2015; Newman et al., 2019, 2020, 2021). Since the conglomeratic beds of the Planteryggen and Plantekløfta formations are advocated by Piepjohn and Dallmann (2014) to reflect the onset of Ellesmerian tectonism, this would therefore imply an abrupt switch in plate tectonic movements and stresses at exactly 383 Ma, i.e., completed within one million year maximum. In addition, mid Famennian–Upper Mississippian sedimentary rocks of the Billefjorden Group and Pennsylvanian–lower Permian rocks of the Gipsdalen Group, which overlie the Andrée Land Group in central–northern Spitsbergen, are believed to have been deposited in extensional basins (Cutbill et al., 1976; Aakvik, 1981; Gjelberg, 1984; Braathen et al., 2011; Smyrak-Sikora et al., 2018). This implies another rapid reversal in regional plate tectonics movements from contraction to extension at ca. 365 Ma. Since regional plate tectonics reorganization and tectonic stress reorientation are known to be relatively slow and gradual processes, such abrupts switches are regarded as highly unlikely. Considering the extensional setting inferred in both the Early–Middle Devonian (Chorowicz, 1992; Piepjohn, 2000; Roy, 2007, 2009; Braathen et al., 2018, 2020; Dallmann and Piepjohn, 2020; Roy et al., unpublished) and mid Famennian–lower Permian (Cutbill et al., 1976; Aakvik, 1981; Gjelberg, 1984; Braathen et al., 2011; Smyrak-Sikora et al., 2018), it is more likely that Ellesmerian contraction never occurred in Svalbard and that the area was subjected to continuous extension throughout the Devonian–Carboniferous. This is also supported by late Silurian–Late Devonian extensional detachment faulting and folding at 430–368 Ma in northwestern Spitsbergen (Braathen et al., 2018) and in the Middle–Late Devonian in northern Spitsbergen (Chorowicz, 1992; Roy, 2007, 2009; Roy et al., unpublished).

The 383–365 Ma estimate for tentative Ellesmerian deformation in shallow-crustal Lower to lowermost Upper Devonian sedimentary rocks in central–northern Spitsbergen partly overlaps with the timing of deep-crustal, 373–355 Ma, amphibolite facies metamorphism in Prins Karls Forland (Majka and Kośmińska, 2017; Faehnrich et al., 2017; Schneider et al., 2018; Kośmińska et al., 2020) and thermal events in Oscar II Land at 377–326 Ma (Michalski et al., 2017). However, the 383–365 Ma estimate reflects the age of stratigraphy in central–northern Spitsbergen, not the age of any specific Ellesmerian structure. In addition, due to conflicting lines of evidence (e.g., postulated prograde metamorphism associated with normal sense of shear), the nature of tectonic stresses during tectonothermal events in Prins Karls Forland and Oscar II Land remains debatable.





Paleomagnetic and $^{40}$Ar–$^{39}$Ar geochronological data from Michalski et al. (2017) do not support a pre-Caledonian link or proximity between the Pearya terrane and western Spitsbergen. On the same trend, detrital zircons in western and central Spitsbergen show affinities with northern Baltica rather than Laurentia in the Paleozoic (Gasser and Andresen, 2013). This suggests that western and central Spitsbergen were located away from the main Ellesmerian belt in northern Greenland and Arctic Canada and, thus, may have escaped Ellesmerian tectonism. This is further supported by the recent discovery of several kilometers thick, thousands of kilometers long, late Neoproterozoic thrust systems crosscutting the whole Barents Sea and the Svalbard Archipelago, thus suggesting that the Svalbard Archipelago was already accreted and attached to Baltica in the late Neoproterozoic (Koehl, 2020b; Koehl et al., 2022).

**Conclusion**

There should be no debate as to the age of the Mimerdalen Subgroup and Billefjorden Group. These are respectively upper Givetian–lower Frasnian (ca. 385–380 Ma) and mid Famennian–Upper Mississippian (ca. 365–325 Ma). The single palynomorph specimen that was not in line with these ages was found in the Mimerdalen Subgroup is a clear misidentification of *Retispora lepidophyta*. Thus, the timing of Ellesmerian tectonism in central–northern Spitsbergen is constrained to 383–365 Ma. Nonetheless, because of the strong impact of Eurekan strain partitioning and extensional detachment-related folding and faulting, much is left to do to quantify the impact, extent and timing of Ellesmerian tectonism in this area (if it ever occurred). Future studies should focus on geochronological dating of presumed Ellesmerian thrusts.

There is also no debate either about the age of the Adriabukta Formation in southern Spitsbergen. This formation is Middle Mississippian in age and is therefore a time-equivalent of the undeformed, sandstone-rich Hornsundeneset Formation. Hence, folding in the Adriabukta Formation is entirely and exclusively ascribed to Eurekan tectonism and the tight character of folding to strain partitioning in the early Cenozoic. Due to lack of robust minimum time constraints, the occurrence of Ellesmerian tectonism in southern Spitsbergen is highly doubtful. Future studies could, if feasible, focus on establishing clear tectonic and stratigraphic relationships in Røkensåta.

Postulated prograde amphibolite-facies metamorphism at 373–355 Ma in pre-Caledonian basement rocks in Prins Karls Forland occurred at a depth of c. 15 kilometers and, thus, has no bearings on the nature of tectonic stress and associated deformation in shallow-crustal Devonian–



Mississippian sedimentary rocks. Top-SW to top-NW normal sense of shear along the dated shear
zone suggests that this episode of postulated prograde metamorphism may actually be related to
shallow-crustal, extensional collapse processes, possibly reflecting progressive burial and
movements along the shear zone during the deposition of collapse sediments. Similar processes are
well documented on the conjugate margin of Svalbard in northeastern Greenland, and in
northwestern Spitsbergen, and these processes involve deep, late Caledonian, high-pressure
metamorphism and shallow-crustal extensional detachments.

Considering the dominantly extensional tectonic settings inferred for shallow-crustal rocks
in late Silurian to early Permian times and the multiple inconsistencies and contradicting lines of
evidence associated to the Ellesmerian Orogeny throughout Svalbard, the accretion of Svalbard to
Baltica as early as the late Neoproterozoic, and the two abrupt and rapid switches in tectonic stress
orientation required in the Late Devonian to account for Ellesmerian tectonism, it is much more
likely that the whole archipelago was subjected to continuous extension from the late Silurian to
early Permian times and escaped Ellesmerian deformation.

**Author contributions**

JBPK wrote the manuscript and designed the figures (contribution: 50%). John E. A.
Marshall and Gilda M. Lopes provided critical input and corrections to the manuscript and
figures (contritution: 25% each).

**Competing interests**

The authors declare that they have no conflict of interest.

**Acknowledgements**

The present study is part of the ARCEx (Research Centre for Arctic Petroleum
Exploration), SEAMSTRESS, and CEED projects (Centre for Earth Evolution and Dynamics). The
author would like to thank all the persons from these institutions that are involved in this project.

**Financial support**



The present study was financed by the ARCEx (grant number 228107), SEAMSTRESS (grant number 287865), and CEED projects (grant number 223272) funded by the Research
Council of Norway, the Tromsø Research Foundation, and six industry partners.

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





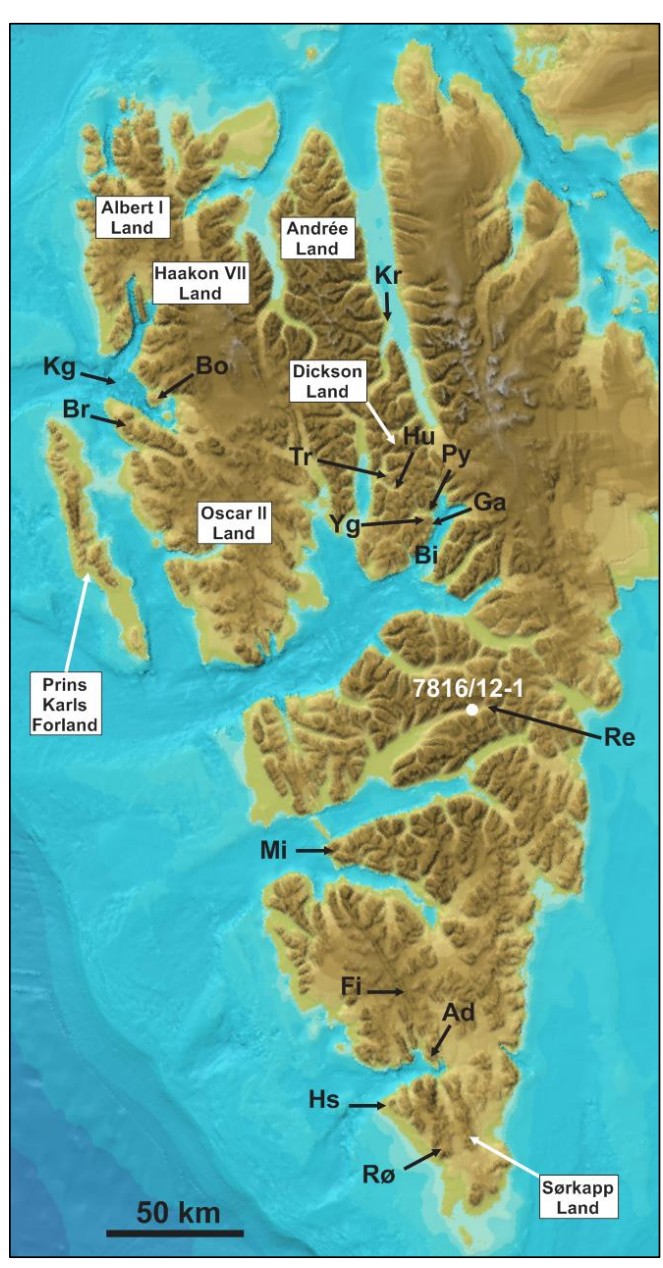


**Figure 1:** Topographic–bathymetric map around Spitsbergen modified after Jakobsson et al. (2012). The location of exploration well 7816/12-1 is shown in white. Abbreviations: Ad: Adriabukta; Bi: Billefjorden; Bo: Blomstrandhalvøya; Br: Brøggerhalvøya; Fi: Fiskeknatten; Ga: Garmdalen; Hs: Hornsundneset; Hu: Hugindalen; Kg: Kongsfjorden; Kr: Krosspynten; Mi: Midterhuken; Re: Reindalspasset; Rø: Røkensåta; Tr: Triungen; Yg: Yggdrasilkampen.








**Figure 2: Late Paleozoic stratigraphic chart of the areas discussed in the text.**