# Peer review of "The timing of the Svalbardian Orogeny in Svalbard: A review"

_Solid Earth, 2022_

## Referee Comment (RC1)

[referee-annotated manuscript omitted]

---

## Referee Comment (RC2)

[referee-annotated manuscript omitted]

---

## Author Response (AR3)

**Reply to Michael Newman**

Dear Dr. Newman,

thank you very much for your input on the manuscript, it is highly appreciated. Here is our reply to your comments. We hope the changes we implemented improve the shortcomings of the manuscript highlighted by your comments and suggestions. Please do not hesitate to contact us shall this not be the case for some comments.

**1. Comments from Dr. Newman**

Comment 1: I found this manuscript thoroughly research and referenced. It is probably a good time to review the Ellesmirian situation in Svalbard as a lot of new data has been published recently. I have attached a pdf that mostly just highlights minor points.

Comment 2: One thing I do think needs to be addressed is the abstract, as it does not really present the results found in the conclusions and elsewhere. A lot of readers do not get further than the abstract, so you need to get their attention by telling them your conclusions, such as, the Mimerdalen Subgroup is upper Givetian to lower Frasnian, etc.

Comment 3: The other thing is more of a suggestion rather than a criticism, in that there is a lot of hyphen use. I think the English might flow a little better if 'to' and 'and', etc., were used when appropriate. That's it really.

Comment 4: line 17: And yet in the conclusions you say it might not have happened in Svalbard at all? The abstract should be more of a synopsis of your results and conclusions rather than a list of the problems with dating the Ellesmerian, etc.

Comment 5: line 43: It might be an idea to put a date range and a general reference for the Caledonian orogeny for those not so familiar with arctic geology.

Comment 6: line 143: So is the specimen considered lost in a ICZN sense? I wonder if it might be worth using Schweitzer's figure of the Spitsbergen specimen compared with a genuine R. lepidophyta at the same scale to illustrate the size difference. Really put a nail in the coffin of the argument. It's up to you.

Comment 7: line 231: Best use 'Ellesmerian' rather than 'Svalbardian' even if they mean the same thing as you have for the rest of the text.

Comment 8: line 235: Did you mean discussion and debate?

Comment 9: line 244: 'Speculation and debate' maybe?

Comment 10: line 282: northwestern or western?

Comment 11: line 796: doi number needed, I found it via the title.

Comment 12: line 799: Similar comment to the one above, only I could not find this one.

Comment 13: line 982: How does a reader get access to this - is there a doi number or website?

Comment 14: line 985: How does a reader get access to this?

Comment 15: line 1061: Py not in the figure caption - presumably this is Pyramiden?

**2. Author's reply**

Comment 1: agreed.

Comment 2: agreed.

Comment 3: agreed.

Comment 4: agreed. See response to comment 2 below.

Comment 5: agreed.

Comment 6: agreed.

Comment 7: agreed. However, the anonymous referee (other referee) is based in Canada where the Ellesmerian Orogeny was define and argues that the term "Svalbardian" should be used in Svalbard rather "Ellesmerian". Therefore, we adjusted the whole manuscript to "Svalbardian".

Comment 8: agreed.

Comment 9: agreed.

Comment 10: disagreed. The paragraph deals with southern Spitsbergen.

Comment 11: agreed.

Comment 12: agreed. This contribution is almost ready and will be submitted to Tektonika later this summer, so unfortunately no DOI available yet, but will be adjusted as soon as possible.

Comment 13: agreed.

Comment 14: agreed.

Comment 15: agreed.

**3. Changes implemented**

Comment 1: none commanded by the reviewer's comment.

Comment 2: added "The Mimerdalen Subgroup is upper Givetian to lower Frasnian (ca. 385–380 Ma) in age and the Billefjorden Group is mid Famennian to Upper Mississippian (ca. 365–325 Ma), therefore constraining the Svalbardian event in central and northern Spitsbergen to 383–365 Ma if it ever occurred. The Adriabukta Formation in southern Spitsbergen is Middle Mississippian and, therefore, cannot have been involved in the Svalbardian event, thus suggesting that all the deformation in southern Spitsbergen in early Cenozoic in age and that strain partitioning processes had a major role in localizing deformation in weaker stratigraphic units. The few geochronological age constraints yielding Late Devonian–Mississippian ages in Svalbard may reflect either Svalbardian contraction or extensional processes and are therefore of no use to validate or invalidate the occurrence of the Svalbardian event. On the contrary, the contradicting lines of evidence used to support the occurrence of the Svalbardian event and new regional geophysical studies suggest that Svalbard was subjected to continuous extension from the late Silurian to early Permian times." lines 31–43.

Comment 3: replaced hyphen characters by "to" lines 18, 21, 35, 41, 43, 49, 57, 84, 87, 100, 108, 117, 146, 161, 191, 195, 201, 209, 228, 229, 231, 349, 353, 361, 372, 379, 398, 399, 402, 426, 431, 468, 497, 507, 518, 519, 525, 527, 530, 531, 532, 558, 559, and 578, by "and" lines 25, 54, 81, 120, 165, 189, 198, 223, 230, 237, 242, 339, 389, 422, 447, 481, 490, 491, 492, 501, 508, 520, 536, 541, 561, and 1075, by "and/or" lines 34, 65, and 100, and by a comma line 129.

Comment 4: see response to comment 2.

Comment 5: added "(ca. 460–410 Ma; Horsfield, 1972; Dallmeyer et al., 1990; Johansson et al., 2004, 2005; Faehnrich et al., 2020)" lines 47–48, and Dallmeyer et al. (1990), Faehnrich et al. (2020), Horsfield (1972), and Johansson et al. (2004, 2005) to the reference list.

Comment 6: added a new S1 supplement illustrating the size difference between the misidentified specimen of Brinkmann (1997), Schweitzer (1999), and Piepjohn et al. (2000), and actual specimen of Retispora lepidophyta from detailed studies by Playford (1976) and Maziane et al. (2002). Also changed supplement S1 into supplement S2 to include the new supplement.

Comment 7: changed "Ellesmerian" into "Svalbardian" throughout manuscript lines 1, 22, 24, 26, 33, 45, 51, 59, 64, 71, 76, 80, 88, 90, 94, 100, 106, 114, 115, 120, 124, 126, 128, 130, 144, 153, 163, 164, 186, 232, 239, 240, 283, 288, 304, 337, 338, 342, 356, 358, 366, 368, 376, 379, 385, 387, 393, 420, 424, 435, 446, 451, 458, 460, 471, 480, 489, 491, 500, 501, 502, 513, 526, 532, 538, 558,

561, 562, 569, 584, 586, and 588. Deleted "Ellesmerian" line 19. Added "Svalbardian (and" line 203. Added "in Svalbard" line 527.

Comment 8: rewrote into "debate" line 241.

Comment 9: replaced "discussion" by "speculation and debate" line 243.

Comment 10: none.

Comment 11: added DOI to the reference.

Comment 12: none.

Comment 13: the Ph.D. thesis is available for purshase at various online libraries, and in paper version at the library of the Norwegian Polar Institute.

Comment 14: the submitted (but never published) manuscript is found at the end of the Ph.D. thesis.

Comment 15: added "Py: Pyramiden; " line 1076.

**Additional revisions by the author of the present manuscript**

-deleted "in Svalbard" line 29.

-replaced "Svalbardian" by "Ellesmerian" lines 33–34.

-deleted reference to Piepjohn (2000) line 56.

-added "mostly" line 75.

-added "e.g., " line 115.

-changed "for example" into "Furthermore" line 119.

-added reference to Maher et al. (2022) lines 148, 263, 527, 567–568, and 599, and to the reference list.

-replaced "initiated in" by "occurred during" line 149, and added "initiating" line 150.

-added ", e.g., in pointing out that field studies based on long-distance observation of poorly exposed and inaccessible transects should be given little (if any) credit." lines 181–183.

-added ", and neither does the claim of Piepjohn et al. (2000) that the older Devonian spores found in the sample with the misidentified specimen of *Retispora lepidophyta* were reworked" lines 241–243.

-added "(Birkenmajer, 1964)" line 280.

-added "Previous works (e.g., Kempe et al., 1997) used the strike and vergence of structures in Blomstrandhalvøya todistinguish Eurekan from presumed Svalbardian structures. This argument is not valid because a single tectonic event may very well produce structures with varying vergence

and strikes, e.g., the Eurekan in Svalbard, which resulted in the formation of east-verging structures in western and southwestern Spitsbergen (e.g, Maher et al., 1986; Dallmann et al., 1988, 1993; Andresen et al., 1994) and northeast-verging folds and thrusts in Brøggerhalvøya (e.g., Bergh et al., 2000; Piepjohn et al., 2001). Furthermore, recent regional studies have shown the occurrence of major, WNW–ESE-striking, several to tens of kilometers thick, thousands of kilometers long, inherited Timanian thrust systems extending from northwestern Russia to western Svalbard (Koehl, 2020; Koehl et al., 2022). One of these structures, the NNE-dipping Kongsfjorden–Cowanodden fault zone, extends into Kongsfjorden, where it was reactivated during the Caledonian and Eurekan events as a sinistral-reverse oblique-slip fault, thus partitioning deformation between northern and southern to western Svalbard during those two events and leading to oppositely verging Eurekan thrust across (e.g., west-verging in Andrée Land and Blomstrandhalvøya and east-verging in Røkensåta and Adriabukta and Hornsund) the fault and to bending Eurekan structures in the vicinity of the fault (e.g., in Brøggerhalvøya)." lines 411–426.

-deleted "strongly" line 434.

-added "west-dipping" line 470.

-added "part of" line 492 and deleted "area" line 493.

-changed "lower" into "early" line 557.

-added "and precise" line 583.

**Reply to Keith Dewing**

Dear Dr. Dewing,

thank you very much for your input on the manuscript, it is highly appreciated. Here is our reply to your comments. We hope the changes we implemented improve the shortcomings of the manuscript highlighted by your comments and suggestions. Please do not hesitate to contact us shall this not be the case for some comments.

**1. Comments from Dr. Dewing**

Comment 1: This paper examines the age of formations in Svalbard and the bearing these ages have on timing of the "Ellesmerian Orogeny" in Svalbard.

Comment 2: The paper has a number of issues: The paper is more a critique than a review. Reviews typically contain sections that would help readers who are not familiar with Spitzbergen geology, like a summary of the stratigraphy and tectonic models, a review of the Ellesmerian Orogeny and known timing relationships. This paper jumps straight into the details of stratigraphic names and seems directed at a very narrow specialist group of experts on Svalbard geology. So my first recommendation would be to either change the word 'Review' in the title to 'Critique', or restructure and expand the paper to give a balanced overview of the regional geology and models that have been proposed for both Svalbard geology and the Ellesmerian Orogeny generally.

Comment 3: If this paper is a narrowly focussed critique of interpretations of Svalbard geology, why not stick with the local term 'Svalbardian Orogeny' for the structural event(s) rather than call it Ellesmerian? Ellesmerian type area is in the Canadian Arctic where it is clearly expressed as a (modern) south-vergent fold and thrust belt with a minimum 25-65 km shortening (depending where you are), and a later strike slip component.

Comment 4: If this paper is a narrowly focussed critique of interpretations of Svalbard geology, why not stick with the local term 'Svalbardian Orogeny' for the structural event(s) rather than call it Ellesmerian? Ellesmerian type area is in the Canadian Arctic where it is clearly expressed as a (modern) south-vergent fold and thrust belt with a minimum 25-65 km shortening (depending where you are), and a later strike slip component.

Comment 5: The introduction does not include events discussed later in the text. It deals with the old models but leaves out key points that are drawn on in the conclusions. Make this part more complete - Caledonian (give age range), Early Devonian rifting (?), putative Ellesmerian folding, possible Late Devonian core complexes/collapse.

Comment 6: Early Devonian rifting and Late Devonian core complexes are not in the intro and need to be discussed. Same for younger strata (Triassic) that are discussed later in the text. Add at least a sentence or two in the introduction.

Comment 7: Eurekan deformation seems important enough in your discussion that it deserves a separate paragraph.

Comment 8: There's a discussion in the paragraph centred on line 75 of an area of undisputed Late Devonian deformation. This seems to fade away in the later discussion (ln 215 for example), where there are doubts cast on the existence of Late Devonian deformation at all. Maybe this is due to the discussion being broken into narrow geographic areas, but, somewhere the text needs to deal with what this reader feels is an inconsistency between line 75 and the later parts of the paper. Undisputed Late Devonian deformation should carry a lot of weight in the interpretation, even if Late Devonian structures have been overprinted elsewhere. Same with the discussion in paragraph spanning lines 300-310.

Comment 9: Sure the outcrops are small and the exposure is less than perfect. BUT – this area does show folded Devonian strata and non folded Triassic shale. Suggesting, with no evidence, that Devonian shales are weaker than the Triassic shales so soaked up the Cenozoic deformation is extremely speculative. It would be more parsimonious to say that there was Late Devonian deformation (widely accepted in the literature), than undocumented bedding parallel slip in Triassic strata only

Comment 10: There is a long section on radiometric age dating of high grade metamorphic rocks. But at the end, you conclude that none of the ages really matter because they could be explain by either compression or extension. Thus the whole section seems a bit pointless as it is written. Lots of detail on individual results, but in each case they are suggested to be useless because they don't give a definitive sense of motion, and could be due to a variety of causes. Could the whole section be reorganized and compressed? Maybe start with the idea that age dating of high T metamorphism is useless in the discussion about Svalbardian timing because they could be either due to crustal

thickening or orogenic collapse. Then briefly discuss the results of each study, but keep it short because they don't help anyway.

Comment 11: Plus, the introduction needs to help the reader with the core complex model. It needs to be discussed earlier rather than being introduced here.

Comment 12: he conclusion draws on the multiple inconstancies and contradictory lines of evidence for the 'Ellesmerian Orogeny' in Svalbard to dismiss the orogeny as unlikely. This conclusion is not supported by the evidence presented (esp. given the apparently solid evidence for Late Devonian deformation discussed around lines 75 and 300). Most scientific concepts are surrounded by inconsistencies and contradictions (think of evolution), yet we don't discard a concept just because it is incompletely understood. We work on getting new evidence to refine the models.

Comment 13: This paper doesn't present any new information, it has some speculation regarding strain partitioning, and provides an incomplete summary of the main geological/tectonic events that make it difficult to follow for the reader not completely familiar with the area.

Comment 14: line 2: Critique would be a better word here. A review usually has sections on regional context (i.e., Greenland and Canada), plus a quick overview of tectonic models, previous work, stratigraphy.  This article takes exception with the interpretation of various datasets, but presents none of the things a reader unfamiliar with the geology of Svalbard might expect.

Comment 15: line 31: because this article deals only with Svalbard, why not call the deformation 'Svalbardian'. The Ellesmerian is a real thing elsewhere, so why confuse the casual reader by critiquing and eventually dismissing an event (the Svalbardian) but calling it Ellesmerian?  If there is, as you suggest, no Late Devonian compression on Svalbard, why give it a name that is just fine in other parts of the Arctic?

Comment 16: line 36: Add Thorsteinsson and Tozer, 1970. This seems to be the paper that originally defines the Ellesmerian Orogeny.

Thorsteinsson, R., Tozer, E.T., 1970. Geology of the Arctic Archipelago. In: Douglas, R.J.W., (Ed.), Geology and Economic Minerals of Canada (5th ed.): Geological Survey of Canada, Economic Geology Report, no. 1, p. 547–590.

Comment 17: line 43: Isn't there some extension and rifting in the Early - or Middle Devonian in Svalbard?

Comment 18: line 43: When did the Caledonian end?  Add a phrase to this sentence with the age range of the Caledonian on Svalbard

Comment 19: line 57: Hey, what about the work suggesting widespread Late? Devonian extension. Try looking at Koehl's work on this :)

This intro part deals with the old models but leaves out key points you draw on in the conclusions.  Make this part more complete - Caledonian (age range), Early Devonian rifting (right?), putative Ellesmerian folding, possible Late Devonian core complexes/collapse.

Comment 20: line 57: how long is 'shortly'?

Comment 21: line 61: maybe expand a sentence or two - what about the Triassic that you mention later.

Then put the Eurekan in a differnt paragraph?

Comment 22: line 73: original unmodified ?

Comment 23: line 101: on Svalbard.

Comment 24: line 109: If you are discussing Ellesmerian deformation, why not talk about timing constraints from Greenland and Canada.  If you are only interested in Svalbard, then perhaps Ellesmerian is the wrong term to use and you should stick with Svalbardian instead?  Using Ellesmerian implied a single, unfied event from Svalbard-Greenland-Canada.

Comment 25: line 172: Do we need all the names here? Can you just say 30 samples have Famennian spore assemblages (Marshall et al. 2015).

Comment 26: line 189: I agree with this in the narrow sense that if that spore is misidentified the age constraint is removed.  BUT, Devonian spores are notoriously recycled. Look at John Utting's work on the Sverdrup Basin. Many Carboniferous samples are dominated by Devonian spores, which are notoriously tough.

Comment 27: line 217: But there is an area of undisputed Late Devonian deformation?  See paragraph centred on line 75.  Add something here to help the reader. Is the deformation mentioned around ln 75 definitively Late Devonian or not?

Comment 28: line 251: didn't the Bergh paper discuss an angular unconformity above the Adriabukta Fm as par of their reasoning for a Devonian age?  If this is a review, then you should at least mention this observation and explain it.

Comment 29: 301: because they are extensively.

Comment 30: line 310: This paragraph is confusing. Sure the outcrops are small and the exposure is less than perfect. BUT - it does show folded Devonian and non folded Triassic shale. I don't see how the argument about strain partitioning holds? Are you trying to say that the Devonian shales are weaker than the Triassic shales so soaked up the Cenozoic deformation? That is not clear from the text and is extremely speculative.

Comment 31: line 324: Okay, you have qualified this a bit, but it does not seem the most parsimonious explanation. There could be Svalbardian deformation (widely accepted), or there is folding in the Devonian but bedding parallel slip in the Triassic, but which can't be documented. I find this argument weak.

Comment 32: line 342: Arguing from the consensus viewpoint seems out of character here! If you are tearing up the received interpretation, why accept it here?

Comment 33: line 365: The structural argument is the stronger one here. The conodonts may be poorly preserved so that they can't be assigned to a particular biozone, but the overall upper Paleozoic character might still be clear.

Comment 34: line 406: This paragraph doesn't offer much and seems out of place with the preceding and following paragraphs.

Comment 35: This seems counter to the linkage you made several paragraphs ago about the linkage between shallower extensional faults and timing of metamorphism? The paragraph seems like a cop out - if you don't like my arguments above, then the data just don't matter anyway.

Comment 36: line 427: woah - that's a jump. Can you add a sentence as to why Ziemniak thought that Barnes was wrong? Re-interpretation or different sample set?

Comment 37: line 433: provided evidence of.

Comment 38: line 445: None of these ages seem useful then if they can be explained by either compression or extension. The whole section seems a bit pointless as it is written. Lots of detail on individual results, but in each case they are suggested to be useless because they don't give a definitive sense of motion, and could be due to a variety of causes. Could the whole section be reorganized and compressed? Maybe start with the idea that age dating of high T metamorphism is useless in the discussion about Svalbardian timing because they could be either due to crustal thickening or orogenic collapse. Then briefly discuss the results of each study, but keep it short because they don't help anyway.

Comment 39: line 460: Big jump here!

the only definitive record... or The only strong record... or The only record of Ellesmerian tectonism that cannot be easily explained by other processess....

there are lots of 'possible' instances that could have been overprinted by Eurekan.

Comment 40: line 565: there are multiple inconsistencies and contradictions in most scientific concepts (think of evolution...). An idea isn't necessarily wrong because it is incompletely understood.

The switches in stress direction

Comment 41: line 863: Wouldn't the old Mann and Townsend 1989 (Geological Magazine v. 126) be relevant here?

Comment 42: line 1062: Is this column necessary if there are no rocks preserved?

Comment 43: line 1062: vary the line weight a bit? Make the line under 'Central Spitzbergen' thicker?

Comment 44: line 1062: what's this empty box?

Comment 45: line 1062: grey.

Comment 46: line 1062: why does Andrée Land Group cover this unconformity? does it include that empty box?

Comment 47: line 1062: make hiatus grey?

Comment 48: line 1062: I'd vary the font a bit. Make the Periods bold or a bit bigger. Turn Devonian and Mississippian 90 degrees if needed.

Add absolute ages at the stage boundaries to make it easier for the reader.

2. **Author's reply**

Comment 1: agreed.

Comment 2: agreed. Because of the overwhelming amount of evidence against the occurrence of the Svalbardian Orogeny, the present manuscript indeed turns out to be a critique as much as a review. This is largely due to the fact that alternative hypotheses to the Svalbardian Orogeny have been consistently disregarded by supporters of the Svalbardian Orogeny (e.g., Piepjohn, 2000, which does not even cite the competing work by Chorowicz, 1992 – this is also the case of the review work by Dallmann and Piepjohn, 2020). Crucial points of emphasis are that (1) research in

the Arctic is relatively expensive, and (2) that research teams pursuing alternative hypotheses (e.g., extensional collapse) ran out of funding (Chorowicz, pers. comm. 2021).

A very (and it cannot be stressed enough) very important point of the paper is that it is directly addressed to specialists of the geology of Svalbard. This is in great part due to the fact that the main author of the manuscript (re-) discovered a large number of old manuscripts on the topic that were not digitized (i.e., not accessible to the wide public and to most researchers) at the library of the Norwegian Polar Institute in Fall 2021. Since the only metric that actually matters to the authors of the present manuscript is that the manuscript is to be read by the appropriate people (and not by the largest number of researchers/scientists/others), it is of absolutely no consequence that the manuscript jumps straight into detailed stratigraphy and terms. The manuscript is addressed to people that are experts on and/or are deeply interested in the topic (this is in agreement with the policy of ResearchGate to delete the "RG Score" and to focus on "Research interest" metric). The time of fellow researchers/scientists/others (i.e., time, money, and human resources) is of great importance to the authors of the present manuscript. It is important that these resources are not wasted in reading a manuscript that is not absolutely necessary for their work.

Comment 3: agreed about the Svalbard part of the deformation.

Comment 4: agreed. See response to comment 3.

Comment 5: agreed. However, Svalbardian folds are already mentioned lines 61–67.

Comment 6: agreed. See also response to comment 5.

Comment 7: agreed.

Comment 8: disagreed. These structures are not undisputed. Most of them are invisible in the field or are obvious misinterpretations. This statement is based on the initial work by Stensiö (1918), Vogt (1938), and Friend (1960) and our own mapping through three field seasons there. Both these older works and ours are consistent with one another and show no major faults in the Devonian succession in Mimerdalen. When considering the stratigraphy and structures at this locality, and paleontological constraints, the team of Dr. Piepjohn (Michaelsen et al., 1997; Brinkmann, 1997; Piepjohn et al., 1997; Michalesen, 1998; Piepjohn, 2000; Piepjohn et al., 2000) is the only team whose findings and mapping differ significantly from all other works. The consistency displayed by all older works and our own strongly suggests that the Svalbardian structures interpreted in this area by Dr. Piepjohn and his group arise from their confusion of the stratigraphy and lack of understanding of the geometry of the Mimerelva Syncline, which curves into an E–W trend in the

north at the corner of Muninelva. As Berry and Marshall (2015) and many other works listed in the present contribution show, the paleontological arguments used by the research group of Dr. Piepjohn was erroneous. Our ongoing manuscripts will show that their interpretation of numerous Devonian structures and of the stratigraphy in the area is incorrect. Notably, the interpretation of Svabardian structures by Dr. Piepjohn and his team does not reconcile several aspects of the local geology, which they did not include in their discussions and considerations (e.g., major drops in the elevation/altitude of the base of the Permian strata overlying the Devonian successions across major valleys in the area). Their interpretation is therefore questionable and the presence of Devonian contraction structures in the area is highly doubtful.

Regarding structures in Blomstrandhalvøya, they are discussed in the present manuscript.

Regarding the paragraph lines 300–310, the interpretation by Dallmann (1992) and Dallmann and Piepjohn (2020) of folded Devonian strata truncated by undeformed Triassic shales is based on long-distance observations and the outcrops are not accessible for detailed inspection because they are located on steep cliffs (Dallmann, pers. comm. 2020). Based on numerous works by other research groups (including Dr. Dallmann), Triassic shales in western Svalbard are well known to have localized Eurekan décollements throughout Svalbard (e.g., Maher, 1984; Maher et al., 1986, 1989; Andresen et al., 1988; Bergh and Andresen, 1990; Haremo and Andresen, 1992; Andresen et al., 1992; Dallmann et al., 1993; Bergh et al., 1997). Since it is not possible to access the boundary between Triassic and Devonian strata there, one cannot prove that there is a décollement, but one cannot prove either that there isn't one. Based on the overwhelming evidence of Eurekan décollements in Triassic shales throughout Svalbard, on our arguments regarding the geometry of the fold (upwards dying-out), and on the poor quality of the outcrop itself (almost completely eroded and mostly made up with loose material) and its inaccessibility for detailed field inspection, we argue that very little credit (if any) should be given to any interpretation of this outcrop (which therefore cannot be used to support the occurrence of the Svalbardian event).

Nevertheless, it is clear from the reviewer's comment that we should be more specific in our argumentation. It should also be proposed that clear methodologies be implemented by geologists in the future to clearly state the uncertainty associated to one's model/interpretation (e.g., poor outcrop quality, inaccessibility, lack of field photograph to document one's claim – e.g., work by Piepjohn et al., 1997; Michaelsen et al., 1997; Piepjohn, 2000). The consistent lack of field photographs of Svalbardian faults in Svalbard in manuscript by the research group of Dr. Piepjohn

is also something to take into account when proving their claims (see following studies: Piepjohn et al., 1997; Michaelsen et al., 1997; Kempe et al., 1997; Piepjohn, 2000).

Comment 9: disagreed. No, we do not know for sure whether the Triassic strata are undeformed due to the poor quality and the inaccessibility of the outcrop. We do not suggest the presence of a bedding-parallel décollement without evidence. We do it based on the works by numerous studies in adjacent Triassic rocks in western Spitsbergen (e.g., Maher, 1984; Maher et al., 1986, 1989; Andresen et al., 1988; Bergh and Andresen, 1990; Haremo and Andresen, 1992; Andresen et al., 1992; Dallmann et al., 1993; Bergh et al., 1997). The supporters of the Svalbardian event did not discuss the possibility of a décollement within Triassic shale in an area (Røkensåta) located within the core of the West Spitsbergen Fold-and-Thrust Belt, although this is a fairly common process as shown by the wealth of examples in the literature (see references above). Instead, they directly inferred the occurrence of the Svalbardian event based on a long-distance observation of a poorly exposed and inaccessible mountain transect.

Comment 10: partly agreed. However, extensive support was drawn by supporters of the Svalbardian event from new geochronological works in favor of its occurrence. It is therefore crucial to review them carefully to show the reader that these actually are inconclusive and do not necessarily support the Svalbardian event since extension is an equally likely scenario. We find the suggestion to start the section about geochronological constraints by stating our conclusion (i.e., that the age do not help in distinguishing contractional from extensional events) not ideal because the manuscript would certainly look less impartial. Nevertheless, we are open to reorganize as suggested by Dr. Dewing should it be also preferred by the other reviewer and by the editor.

Comment 11: agreed. See response to comment 5.

Comment 12: we agree with Dr. Dewing in that "Most scientific concepts are surrounded by inconsistencies and contradictions", and that the Svalbardian event (if it ever occurred) "is incompletely understood". As it turns out, evidence initially (> 30 years ago) thought to be undisputably in favor of the Svalbardian event are highly questionable by modern research standards and ethics (e.g., they are not reproducible and not replicable). See also response to comment 8. It is therefore important that scientists focus again on the Svalbardian event and also on possible (long-forgotten) alternatives (e.g., Chorowicz, 1992; Roy, 2007, 2009, who ran out of fundings earlier than the research group of Dr. Piepjohn and therefore did not manage to further develop their realistic extensional alternative to the Svalbardian event – Chorowicz, pers. comm.

2019; note that Chorwicz, 1992 and Roy, 2007, 2009 do discuss the Svalbardian event before dismissing it and arguing in favor of an extensional origin, whereas Dr. Piepjohn and his team do not consider any other alternative in their works, e.g., Piepjohn et al., 1997, 2000, Michaelsen et al., 1997, Michaelsen, 1998, and Piepjohn, 2000; Piepjohn and Dallmann, 2014). The authors of the present manuscript do not aim at rejecting firmly the occurrence of the Svalbardian event, but at re-establishing a balance in the literature, which is, up to present day, strongly unbalanced/biased towards the Svalbardian model. Noteworthy, a model for continuous late Silurian–Carboniferous extension is gaining momentum thanks to the work by Braathen et al., (2018, 2020) and Maher et al. (2022).

Comment 13: the authors of the present manuscript concede that the present manuscript does not present any new data and as such is listed as a review manuscript. We also agree that the introduction section was incomplete (see responses to comments 5, 6, 7, 8, and 11). However, we disagree in that the present manuscript "has some speculation regarding strain partitioning". The present manuscript describes many uncertainties that supporters of the Svalbardian event should have described in their manuscripts, such as the fact that many observations were made from great distances (Dallmann, pers. comm. 2020) on incomplete mountain transects, which are partly to mostly made up with loose material and which are inaccessible for detailed inspection because located on steep cliffs, and that most structural measurements were taken within riverbeds, i.e., not directly at the contact of Devonian rocks and overlying successions. In the very few places where this contact is exposed and accessible for detailed field inspection, the outcrops either show gently tilted (i.e., undeformed) Devonian strata under undeformed post-Devonian strata, or bedding-parallel décollements within coal- and shale-rich units such as the Billefjorden Group (e.g., Koehl, 2021), thus strongly supporting Eurekan strain partitioning and shielding of post-Devonian strata from Eurekan deformation by weak sedimentary layers, which absorbed most of the deformation. Note that strain partitioning is also clearly observed on 2D seismic transects (i.e., continuous data in two dimensions) in Svalbard, e.g., at the base of the Billefjorden Group in Sassenfjorden (Koehl, 2021) and between strongly deformed Devonian and mildly deformed Permian carbonates in Billefjorden (Koehl et al., submitted), thus further supporting the important role played by Eurekan strain partitioning in Spitsbergen. Again, we do not firmly reject the occurrence of the Svalbardian event, but seriously question the evidence and methods used to collect the evidence supporting its

occurrence. It is up to the scientific community to collect more interdisciplinary evidence with modern and ethical methods to further validate or reject its occurrence.

Comment 14: agreed. "Critique" might also be a suitable term for this review manuscript, but it is mostly because the manuscript calls the reader's attention on the numerous biases and weaknesses involved in the Svalbardian event model. We also agree that some information was missing in the introduction section (see also responses to comments 5, 6, 7, 8, 11, and 13).

Comment 15: agreed. See response to comment 3.

Comment 16: agreed.

Comment 17: agreed. Late–post-Caledonian collapse extension occurred in Svalbard in the latest Silurian to Devonian (Carboniferous?). See response to comment 5.

Comment 18: agreed. See response to comment 5.

Comment 19: agreed. Also see response to comment 5.

Comment 20: agreed.

Comment 21: agreed, though mention of the Triassic rocks was added in the previous paragraph and the Eurekan event was ascribed a full paragraph three paragraphs below. See responses to comments 6, 7, and 8.

Comment 22: agreed.

Comment 23: agreed. However, we updated the event's name to "Svalbardian" (see response to comment 3). "in Svalbard" is therefore not needed anymore.

Comment 24: agreed. See response to comment 3.

Comment 25: disagreed. Stating all the 12 characteristic species names found in each of the 30 samples adds contrast to the work by Piepjohn et al. (2000) who based their interpretation on one specimen of (misidentified) Retispora lepidophyta in only one sample.

Comment 26: agreed. The sentence needs rewriting to be more accurate.

Comment 27: no, it is not necessarily Devonian. See responses to comments 8, 9, and 12.

Comment 28: yes, indeed, Bergh et al. (2011) mention an unconformity between the Adriabukta Formation and overlying Pennsylvanian rocks. They cite Dallmann (1992) who cites Birkenmajer (1964, 1975) pp. 53. Dallmann (1992) himself recognizes in the next paragraph (pp. 53) that "There is, however, no doubt about the presence of the angular unconformity. This fact by itself would not ascertain the existence of a folding event" (i.e., in the Devonian). He further argues that the unconformity needs to be combined to observations made in Røkensåta to infer a Late

Devonian event of contractional folding. As mentioned in the present manuscript (last two paragraphs of the same section, in the third sub-section) and in our responses to comments 8 and 9, observations made from great distance at the Røkensåta locality should not be given any credit due to the extremely poor quality of the outcrop section there and the impossibility to inspect the contact between Devonian and Triassic rocks in detail (i.e., actually step on it) because it is located on steep cliffs. As mentioned in Koehl (2020a) and in Supplement S2 in the present manuscript, the unconformity between the Adriabutka and Hyrnefjellet formations in Adriabukta may, as well, be explained by extensional processes, such as core complex exhumation. This is mentioned in the third sub-section (first paragraph) of the same section.

Comment 29: agreed.

Comment 30: on the contrary, we argue that the Triassic shales are much weaker than the Devonian shales and that the former soaked up Eurekan deformation. This is documented in many places in western Svalbard where the Triassic shales accommodated large amounts of deformation and displacement in décollement levels (Maher, 1984; Maher et al., 1986, 1989; Andresen et al., 1988; Bergh and Andresen, 1990; Haremo and Andresen, 1992; Andresen et al., 1992; Dallmann et al., 1993; Bergh et al., 1997). As a result, deformation within Triassic shales may occur at much lower scale (e.g., meter-scale beds) than in less weak Devonian shales (meso- to macro-scale folds). This is the main argument behind inspecting the contact between the Devonian and the Triassic in detail and not from great distance. We agree that this should be specified in the paragraph mentioned.

Comment 31: disagreed. The same can be said about Svalbardian deformation: it can't be documented because the folds/deformation there have/has not been dated yet, and, most importantly, the contact between the Devonian and Triassic has not been inspected in detail, therefore casting strong doubts on the Svalbardian hypothesis since the research community has widely demonstrated in many instances in Svalbard (Maher, 1984; Maher et al., 1986, 1989; Andresen et al., 1988; Bergh and Andresen, 1990; Haremo and Andresen, 1992; Andresen et al., 1992; Dallmann et al., 1993; Bergh et al., 1997; Koehl, 2021), but also worldwide, the effect of thin décollement levels even at smaller scale (meter scale, i.e., undetectable from great distance; e.g., in Pyramiden – Koehl, 2021). As mentioned in our response to comment 30 (see section about changes implemented to the manuscript), it may be possible to inspect the contact with a drone and, therefore, to document the presence of bedding-parallel décollement if the whole base of the Triassic succession is exposed and if the exposures are of reasonable quality, which may very well

not be the case in Røkensåta judging from the amount of collapsed/loose material on the long distance photographs by Dallmann (1992) and Dallmann and Piepjohn (2020). Triassic bedding-parallel décollements of early Cenozoic age have been documented in many places along the west coast of Svalbard, and this is much more widely accepted and by many research groups (Maher, 1984; Maher et al., 1986, 1989; Andresen et al., 1988; Bergh and Andresen, 1990; Haremo and Andresen, 1992; Andresen et al., 1992; Dallmann et al., 1993; Bergh et al., 1997) than the Svalbardian event, which was mostly proposed by one research group in the 90s with arguments that are at the very least doubtful and need re-examination, as we show in the present manuscript and, e.g., in Koehl (2020a, 2021). Further work is needed before we can firmly reject or validate the Svalbardian event, and the scientific community needs to be aware of this especially because it seems that many researchers think that the Svalbardian event is widely and firmly accepted.

Comment 32: the authors of the present manuscript are a little uncertain about what is meant by the reviewer's comment. This paragraphs aims at showing the reader that some structures in Blomstrandhalvøya were already ascribed an early Cenozoic age and that the arguments used to distinguish between Svalbardian and Eurekan folds and thrusts in the area are extremely weak. Since the initial publication by Kempe et al. (1997) is written in German and is not available at any online repository, its content is not accessible to most researchers (but it is accessible to the authors of the present manuscript). Thus far, this work was only accessible through reading discussions in subsequent papers by the same research group (e.g., Piepjohn, 2000). It is therefore of great importance to state the information contained in this manuscript clearly. We are not tearing up the interpretation/speculation of Kempe et al. (1997) that NW-verging thrusts are early Cenozoic in age. The authors of the present manuscript await further clarification by the reviewer if they misunderstood the comment.

Comment 33: agreed, the structural argument presented in the previous paragraph is stronger than the one presented in the present paragraph. Does the reviewer suggest deletion/shortening (other?) of the present paragraph?

Comment 34: agreed. However, the paragraph discusses the reliability of the ages presented in Kosminska et al. (2020) and is therefore relatively important.

Comment 35: agreed. This is to show that no matter how one looks at it, arguments supporting Svalbardian contraction in western Svalbard are tentative at best. The author's of the present manuscript are aware of the "dilution effect" weak arguments may have on strong arguments

(https://www.ted.com/talks/niro_sivanathan_the_counterintuitive_way_to_be_more_persuasive?language=en). However, authors of the present manuscript thought it best to present all the arguments at hand so that future works may assess accurately the reliability of this claim and of previous works.

Comment 36: agreed.

Comment 37: agreed.

Comment 38: to the authors of the present manuscript's knowledge, it is the first time that the geochronological arguments used to support the occurrence of the Svalbardian event are reviewed in detail. Shortening the section is therefore not an ideal option. The authors of the present manuscript agree that reorganizing the section starting "with the idea that age dating of high T metamorphism is useless in the discussion about Svalbardian timing because they could be either due to crustal thickening or orogenic collapse" and "then briefly discuss the results of each study" is judicious.

Comment 39: agreed. The Røkensåta locality is the only potential instance of Late Devonian contraction left in Spitsbergen since all the other arguments supporting Svalbardian tectonism are not valid. However, it is speculative to say that Eurekan structures overprint Svalbardian structures everywhere. What about Timanian and Caledonian structures, which are very weakly (if at all) overprinted by early Cenozoic structures? We concede that Timanian structures are extensively overprinted by Caledonian structures, but the initial Timanian signal is still preserved in smaller shear zones reworked during the Caledonian Orogeny (e.g., Faehnrich et al., 2020). Thus, should the Svalbardian event have occurred in Svalbard, one should expect to obtain Late Devonian ages for structures showing unquestionable contractional kinematics. This is not the case yet.

Comment 40: see response to comment 12.

Comment 41: partly agreed, but the mentioned reference would not add much to the discussion.

Comment 42: yes, it is because the authors of the present manuscript intend to show the stratigraphic column at each major locality mentioned in the text in order to help the reader understand the geological setting in each area.

Comment 43: agreed.

Comment 44: agreed, this is a mistake.

Comment 45: agreed.

Comment 46: agreed. See response to comment 44.

Comment 47: agreed. See response to comment 45.

Comment 48: agreed.

**3. Changes implemented**

Comment 1: none commanded by the reviewer's comments.

Comment 2: none.

Comment 3: changed "Ellesmerian" into "Svalbardian" throughout manuscript lines 1, 22, 24, 26, 33, 45, 51, 59, 64, 71, 76, 80, 88, 90, 94, 100, 106, 114, 115, 120, 124, 126, 128, 130, 144, 153, 163, 164, 186, 232, 239, 240, 283, 288, 304, 337, 338, 342, 356, 358, 366, 368, 376, 379, 385, 387, 393, 420, 424, 435, 446, 451, 458, 460, 471, 480, 489, 491, 500, 501, 502, 513, 526, 532, 538, 558, 561, 562, 569, 584, 586, and 588. Deleted "Ellesmerian" line 19. Added "Svalbardian (and" line 203. Added "in Svalbard" line 527.

Comment 4: see response to comment 3.

Comment 5: added "(ca. 460–410 Ma; Horsfield, 1972; Dallmeyer et al., 1990; Johansson et al., 2004, 2005; Faehnrich et al., 2020)" lines 47–48, and Dallmeyer et al. (1990), Faehnrich et al. (2020), Horsfield (1972), and Johansson et al. (2004, 2005) to the reference list. Added "and subsequent deposition of thick upper Silurian–Devonian sedimentary successions during late–post-orogenic collapse (Gee and Moody-Stuart, 1966; Friend et al., 1966; Friend and Moody-Stuart, 1972; Murascov and Mokin, 1979; Manby and Lyberis, 1992; Manby et al., 1994; Friend et al., 1997; McCann, 2000; Dallmann and Piepjohn, 2020)" lines 48–51. Added "In addition, new geochronological and structural work in northern Svalbard shows that collapse-related extension leading to the exhumation of the Bockfjorden Anticline as a core complex lasted from the late Silurian to the Late Devonian (Famennian at 368.42 ± 0.81 Ma; Braathen et al., 2018), i.e, possibly overlapping with Svalbardian contraction." Lines 57–60. Added "and understand its interplay with potentially coeval collapse processes (e.g., Braathen et al., 2018)" lines 114–115.

Comment 6: added "The latter were interpreted to be unconformably covered by presumed undeformed, shale-rich, poorly exposed Triassic strata (Dallmann, 1992)." Lines 67–69. See also response to comment 5.

Comment 7: added "Distinguishing Svalbardian from Eurekan structures is problematic. In Arctic Canada and Greenland, Ellesmerian structures are thought to be overprinted almost everywhere by subsequent Eurekan structures (e.g., Piepjohn et al., 2015). This is also the case to some extent in

Svalbard, where Svalbardian and Eurekan folds and thrusts are believed to both show dominantly east-verging geometries in the south, but opposite vergence in the north where Svalbardian structures display mostly top-west attitudes (Dallmann and Piepjohn, 2020). Another issue arises from the complexity of the Eurekan fold-and-thrust belt throughout Spitsbergen, which involves numerous décollements localized in shale-rich stratigraphic units, such as the Lower Triassic (Maher, 1984; Maher et al., 1986, 1989; Andresen et al., 1988; Bergh and Andresen, 1990; Haremo and Andresen, 1992; Andresen et al., 1992; Dallmann et al., 1993; Bergh et al., 1997)." Lines 108–118, and split the paragraph into two.

Comment 8: added "The structures in the Dickson Land area are actually highly questionable and are addressed in two separate manuscripts (Koehl et al., in prep.; Koehl and Stokmo, in prep.), and will therefore not be reviewed in detail in the present manuscript." lines 96–98. Added "Important uncertainties around this interpretation are discussed for the first time in the present work and suggest that interpretation based on this outcrop should be given little to no credit." lines 69–71. Added "Therefore, we propose that little to no weight should be given to any interpretation of these two poorly exposed and inaccessible outcrops." lines 358–359.

Comment 9: none.

Comment 10: awaiting further instruction from the editor and the reviewers.

Comment 11: see response to comment 5.

Comment 12: none. See also response to comment 8.

Comment 13: see responses to comments 5, 6, 7, 8, and 11.

Comment 14: see responses to comments 5, 6, 7, 8, 11, and 13.

Comment 15: see response to comment 3.

Comment 16: added reference to Thorsteinsson and Tozer (1970) lines 38–39, and to the reference list.

Comment 17: see response to comment 5.

Comment 18: see response to comment 5.

Comment 19: added ", possibly during widespread latest Devonian–Mississippian extension," line 74 and "Koehl and Muñoz-Barrera, 2018" lines 77 and 550–551, and Koehl and Muñoz-Barrera, 2018 to the reference list. Also see response to comment 5.

Comment 20: added "(i.e., immediately up to a few million years after)" line 72.

Comment 21: see responses to comments 6, 7, and 8.

Comment 22: added "(unmodified)" line 91.

Comment 23: see also response to comment 3.

Comment 24: see response to comment 3.

Comment 25: none.

Comment 26: replaced "is no longer valid" by "no longer has any supporting argument in Svalbard" line 224.

Comment 27: see responses to comments 8, 9, and 12.

Comment 28: added "The fact that the shear zone does not seem to crosscut the Hyrnefjellet Formation and instead abruptly dies out at the unconformity (see sketch in figure 5 in Bergh et al., 2011) rather supports a formation as a normal fault in the Mississippian (Supplement S2)." lines 319–322.

Comment 29: added "they were" line 340.

Comment 30: added "weak" line 342, ", which localized large amounts of Eurekan deformation and displacement along décollement levels" lines 343–344, ". Triassic are known to be much weaker than Devonian shales and to have preferentially localized Eurekan deformation at a much lower scale (e.g., décollements with kilometer-scale displacement in the Triassic shales versus open meso- to macro-scale folds with limited to no displacement within Devonian shales)" lines 362–365, and "until further inspection of the contact is made from very close range (e.g., using a drone?)" lines 369–370.

Comment 31: none.

Comment 32: none yet, but awaiting further clarification by the reviewer if needed.

Comment 33: none yet. Awaiting further instruction by the reviewer.

Comment 34: moved the paragraph at the end of the present section and moved "Furthermore" from line 465 to line 472.

Comment 35: none.

Comment 36: changed "; Ziemniak et al., 2020)" into ") by Ziemniak et al. (2020), who obtained comparable ages for the same unit without the 365–344 Ma disturbance, which they attribute to fluid circulation. This is also partly supported by the poorer statistical reliability of the 365–344 Ma ages as documented by Barnes et al. (2020)" lines 486–489. Deleted "therefore" line 490. Changed "is most likely" into "may as well be" line 492.

Comment 37: replaced "evidenced" by "provided evidence of" line 495.

Comment 38: added "However, none of the ages in Prins Karls Forland are of any use in discussing the timing of the Svalbardian event since they could either reflect crustal thickening or late–post-orogenic collapse." lines 431–433 and deleted "However, " line 434. Replaced "In Oscar II Land (location in Figure 1), " by "Geochronological ages in Oscar II Land (location in **Error! Reference source not found.**) are also useless in discussing the timing of the Svalbardian event since they may equally reflect extensional processes." lines 487–488. Added "that" line 489. Deleted "suggesting it potentially reflects Svalbardian deformation. However, these ages" line 490. Replaced "This" by "The 365–344 Ma" line 494.

Comment 39: replaced "possible" by "potential" line 528.

Comment 40: see response to comment 12.

Comment 41: none.

Comment 42: none.

Comment 43: thickened some major lines in the table.

Comment 44: adjusted rectangle in table.

Comment 45: made all hiatuses grey and added "hiatus" to the legend.

Comment 46: see response to comment 44.

Comment 47: see response to comment 45.

Comment 48: added absolute ages to the age scale, turned the font of the age scale bold, and increased the font size of the legend. Also added "The ages in the time scale are in Ma and are from Walker et al. (2018)." to the caption of Figure 2, and reference to Walker et al. (2018) in the reference list.

**Additional revisions by the author of the present manuscript**

-deleted "in Svalbard" line 29.

-replaced "Svalbardian" by "Ellesmerian" lines 33–34.

-deleted reference to Piepjohn (2000) line 56.

-added "mostly" line 75.

-added "e.g., " line 115.

-changed "for example" into "Furthermore" line 119.

-added reference to Maher et al. (2022) lines 148, 263, 527, 567–568, and 599, and to the reference list.

-replaced "initiated in" by "occurred during" line 149, and added "initiating" line 150.

-added ", e.g., in pointing out that field studies based on long-distance observation of poorly exposed and inaccessible transects should be given little (if any) credit." lines 181–183.

-added ", and neither does the claim of Piepjohn et al. (2000) that the older Devonian spores found in the sample with the misidentified specimen of *Retispora lepidophyta* were reworked" lines 241–243.

-added "(Birkenmajer, 1964)" line 280.

-added "Previous works (e.g., Kempe et al., 1997) used the strike and vergence of structures in Blomstrandhalvøya todistinguish Eurekan from presumed Svalbardian structures. This argument is not valid because a single tectonic event may very well produce structures with varying vergence and strikes, e.g., the Eurekan in Svalbard, which resulted in the formation of east-verging structures in western and southwestern Spitsbergen (e.g, Maher et al., 1986; Dallmann et al., 1988, 1993; Andresen et al., 1994) and northeast-verging folds and thrusts in Brøggerhalvøya (e.g., Bergh et al., 2000; Piepjohn et al., 2001). Furthermore, recent regional studies have shown the occurrence of major, WNW–ESE-striking, several to tens of kilometers thick, thousands of kilometers long, inherited Timanian thrust systems extending from northwestern Russia to western Svalbard (Koehl, 2020; Koehl et al., 2022). One of these structures, the NNE-dipping Kongsfjorden–Cowanodden fault zone, extends into Kongsfjorden, where it was reactivated during the Caledonian and Eurekan events as a sinistral-reverse oblique-slip fault, thus partitioning deformation between northern and southern to western Svalbard during those two events and leading to oppositely verging Eurekan thrust across (e.g., west-verging in Andrée Land and Blomstrandhalvøya and east-verging in Røkensåta and Adriabukta and Hornsund) the fault and to bending Eurekan structures in the vicinity of the fault (e.g., in Brøggerhalvøya)." lines 411–426.

-deleted "strongly" line 434.

-added "west-dipping" line 470.

-added "part of" line 492 and deleted "area" line 493.

-changed "lower" into "early" line 557.

-added "and precise" line 583.

**Reply to Silvia Gardin**

Dear Dr. Gardin,

thank you very much for your input on the manuscript, it is highly appreciated. Here is our reply to your comments. We hope the changes we implemented improve the shortcomings of the manuscript highlighted by your comments and suggestions. Please do not hesitate to contact us shall this not be the case for some comments.

**1. Comments from Dr. Gardin**

Comment 1: Maybe because the Paleozoic is not my cup of tea, but I find Keith Dewing's comment n°2 very appropriate: this manuscript is aimed at a very specialized audience and many of the concepts and ideas are like acquired. I'm not criticizing this, but I find it very regrettable that the word (and tool) palynology is not even mentioned in the summary! Your age reconsideration is based almost exclusively on palynological analyses and taxonomy, this aspect should be well detailed in the summary. If the non-specialist will not venture further into the detailed reading of the text, at least he will know, thanks to an informative summary that this work is based on a taxonomic and palynological reinterpretation. You write "the present work revise the age..." but on what this revision is based we only discover in the text! Likewise, in the conclusions, I urge you to make this clearer to have a wider public impact and better fit to the Journal audience.

Comment 2: As for the use of "critique" instead of "review", I find that to be quite subtle, but you also have the choice of other terms as well, such as « re-interpretation » or « reconsideration » which you can use instead one or the other according to your convenience.

**2. Author's reply**

Comment 1: agreed, the authors of the present manuscript are actually shocked that this did not cross anyone's mind at some point in the writing process. It is indeed important to specify the types of constraints used in the present review. The present work does not reinterpret any of the ages obtained by previous studies, but tries to sort out reliable from unreliable constraints.

Comment 2: the authors of the present manuscript did not realize that the comment of Dr. Dewing regarding the term "critique" versus "review" targeted the title of the manuscript and not the registration of the type of manuscript in Solid Earth. The authors of the present manuscript still believe that the term "review" applies since the manuscript reviews all the age constraints available and giving insights in the timing of the Svalbardian event. As it turns out, the review of these ages shows that they overwhelmingly point at major inconsistencies in the timing of the event, possibly that the event did not occur at all. The authors of the present manuscript are open to change the title of the manuscript but feel that the term "critique" would make some readers think before they even start reading that the present study was not conducted in a most objective and impartial way.

**3. Changes implemented**

Comment 1: added "including notably palynological, paleontological, and geochronological evidence. This" lines 25–26, and deleted ", which" line 26. Added "Palynological and paleontological evidence suggest that" line 28, "Palynological ages indicate that" line 31, "and are robustly constrained by palynoligcal and paleontological markers" lines 623–624, and "Palynological evidence confirm that" line 632.

Comment 2: none yet, but could change the term "review" in the title if judged necessary.

[revised manuscript text omitted]

**Supplements**

[Figure]

S1: Comparison of (a–e) typical specimen of *Retispora ledipophyta* from Playford (1976) and Maziane et al. (2002) and (f) the lone specimen misidentified for *Retispora lepidophyta* by Brinkmann (1997), Schweitzer (1999), and Piepjohn et al. (2000). One of the most important discrepancies is the size of the misidentified specimen (c. 250 µm), which compares poorly to the actual size range (27–112 µm) obtained by detailed studies on *Retispora lepidophyta* (Playford, 1976; Maziane et al., 2002). See supplement DR3 in Berry and Marshall (2015) for a complete list of all discrepancies. (a–e) are after Matyja et al. (2020) and (f) is after Brinkmann (1997).

[Figure]

**S1S2: (a)** Interpreted cross-section of the Adriabukta transect (see Fig. 1 for location of Adriabukta) redrawn after Bergh et al. (2011), and (b) restored prior to Mesozoic sedimentation and early Cenozoic Eurekan deformation. Notice the basement fabrics (in the basement lens within the Adriabukta Formation) parallel to bedding surfaces in the Adriabukta Formation and the eastwards dip of the Mariekammen Shear Zone. Abbreviations: EHF: Eastern Hornsund Fault; MSZ: Mariekammen Shear Zone. Illustration is from Koehl (2020a).